# Polynomial Convergence of Bandit No-Regret Dynamics in Congestion Games

## Abstract

We present an online learning algorithm in the bandit feedback model that, once adopted by all agents of a congestion game, results in game-dynamics that converge to an $\epsilon$-approximate Nash Equilibrium in a polynomial number of rounds with respect to $1/\epsilon$, the number of players and the number of available resources. The proposed algorithm also guarantees sublinear regret to any agent adopting it. As a result, our work answers an open question from (Cui et al., 2022) and extends the recent results of (Panageas et al., 2023) to the bandit feedback model. Our algorithm can be implemented in *polynomial time* for the important special case of Network Congestion Games on Directed Acyclic Graphs (DAG) as barycentric spanners can efficiently be constructed in this case. We complete our work by further proposing a natural, exact, 1-barycentric spanner construction for DAGs.

## 1. Introduction

Congestion games represent a class of multi-agent games where $n$ self-interested agents compete over $m$ resources. Each agent chooses a subset of these resources, and their individual costs depend on the utilization of each selected resource (i.e., the number of other agents utilizing the same resource). For instance, in *Network Congestion Games*, a graph is given, and each agent $i$ aims to travel from an initial vertex $s_i$ to a designated destination vertex $t_i$. The agent must then select a set of edges (i.e resources) constituting a valid $(s_i, t_i)$-path in the graph, while also trying to avoid congested edges.

Congestion games have been extensively studied over the years due to their wide-ranging applications (Koutsoupias & Papadimitriou, 1999; Roughgarden & Tardos, 2002; Christodoulou & Koutsoupias, 2005; Fotakis et al., 2005;

de Keijzer et al., 2010; Roughgarden, 2009). They always admit a Nash Equilibrium (NE) which is a *steady state* at which no agent can decrease their cost by unilaterally deviating to another selection of resources (Rosenthal, 1973). A Nash equilibrium is a static solution concept meaning that it does not describe how agents can end up in such an equilibrium state nor it indicates how agents should update their strategies. It is well-known that *better response dynamics*, in which agents sequentially update their resource selection, converges to a Nash Equilibrium and achieves accelerated rates for interesting special cases of congestion games (Chien & Sinclair, 2007; Gairing et al., 2004).

Despite these positive convergence results, *better response dynamics* admit several drawbacks. In case of simultaneous updates by agents, better response dynamics may not converge to NE. Moreover a better response comes with the assumption that the agents are aware of the loads of all the available resources (Chien & Sinclair, 2007). Finally, better response does not come with any kind of guarantees to individual agents, which raises concerns as to why a selfish agent should behave according to best-response.

Fortunately the online learning framework (Hazan, 2019) provides a very concrete answer as to what natural strategic behavior means (Even-Dar et al., 2009). There are various *no-regret* algorithms that a selfish agent can adopt in the context of repeated game-playing in order to guarantee that no matter the actions of the other agents, the agent suffers a cost comparable to the cost of the *best fixed action* (Arora et al., 2012; Zinkevich, 2003) chosen in hindsight. The guarantee holds even under a *bandit feedback model* in which the agent only learns the total cost of its selected actions (resource-selection in the context of congestion games) (Auer et al., 2002; Audibert & Bubeck, 2009). Due to the merits of such no-regret schemes, there exists a long line of research providing no-regret algorithms under bandit feedback in the context of congestion games, which are studied under the name of online routing or linear bandits in the online learning literature (Awerbuch & Kleinberg, 2004; Dani et al., 2007a; György et al., 2007; Bubeck et al., 2012; Cesa-Bianchi & Lugosi, 2012; Kalai & Vempala, 2005; Neu & Bartók, 2013; Audibert et al., 2014).

Despite the long interest in bandit online learning algorithms for congestion games, the convergence to Nash Equilibrium

[1]Anonymous Institution, Anonymous City, Anonymous Region, Anonymous Country. Correspondence to: Anonymous Author <anon.email@domain.com>.

Preliminary work. Under review by the International Conference on Machine Learning (ICML). Do not distribute.

of such bandit no-regret learning algorithms is not as well explored. The broad question under consideration here is whether or not the uncoordinated selfish behavior of agents can converge to equilibrium. In this area, the seminal work of (Blum et al., 2006) studied the context of non-atomic congestion games, i.e., infinitesimal agents, and established that the behavior of *any* no-regret learning algorithm converges in the average sense to a Wardrop equilibrium. The non-atomic setting has the advantage of convex landscapes and the fact that Coarse Correlated and Wardrop equilibria coincide. The same does not hold in atomic games (i.e finite agents).

To the best of our knowledge (Cui et al., 2022) were the first to provide an update rule (performing under bandit feedback) that once adopted by all agents of an atomic congestion game, the resulting strategies converge to an $\epsilon$-approximate Nash Equilibrium with rate polynomial in $n$, $m$ and $1/\epsilon$. However their method does not guarantee the no-regret property. As a result, (Cui et al., 2022) asked the following question:

*Is there a no-regret algorithm, in the bandit feedback model, that once adopted by all agents, results in strategies that converge to an $\epsilon$-approximate Nash Equilibrium in* $\text{poly}(n, m, 1/\epsilon)$ *rounds?*

In their recent work (Panageas et al., 2023) provided a positive answer for the *semi-bandit feedback model* in which the agents learn the cost of every single selected resource. In contrast, under *bandit feedback*, the agents only learns the overall, total sum, cost of the selected resources and thus does not have access to the more granular information accessible in *semi-bandit feedback*.

### 1.1. Our Contribution and Techniques

The main contribution of our work consists in providing a positive answer to the open question of (Cui et al., 2022). More precisely, we provide an online learning algorithm, called *Online Gradient Descent with Caratheodory Exploration* (BGD − CE), that simultaneously provides both regret guarantees and convergence to Nash Equilibrium.

**Informal Theorem** *There exists an online learning algorithm (*BGD − CE*) that performs under bandit feedback and guarantees* $\mathcal{O}(m^{2.8}T^{4/5})$ *regret to any agent that adopts it. Moreover if all agent adopt* BGD − CE*, then the resulting strategies converge to an $\epsilon$-Nash Equilibrium after* $\mathcal{O}(n^{13.5}m^{13}/\epsilon^5)$ *steps.*

Our proposed online learning algorithm additionally improves on the convergence rate of the algorithm of (Cui et al., 2022). The table 1 summarizes the regret bounds and the convergence results of the various online learning algorithms proposed over the years.

*Table 1.* Comparison with previous related work. $^\star$A regret bound of $\mathcal{O}\left(m^3 T^{3/4}\right)$ can be obtained under a different choice of step size and exploration coefficients. (B:Bandit, SB: Semi-Bandit)

| Regret Gurantees and Convergence rates | | | |
|---|---|---|---|
| Method | Regret Guarantees | Convergence to NE | Feedback |
| (Auer et al., 2002) | $\mathcal{O}(\sqrt{2^m T})$ | No | B |
| (Awerbuch & Kleinberg, 2004) | $\mathcal{O}(m^{5/3}T^{2/3})$ | No | B |
| (Dani et al., 2007a) | $\mathcal{O}(m^{1.5}\sqrt{T})$ | No | B |
| (Cui et al., 2022) | Not Available | $\mathcal{O}(n^{11}m^{12}/\epsilon^6)$ | B |
| (Panageas et al., 2023) | $\mathcal{O}(m^2T^{4/5})$ | $\mathcal{O}(n^6m^7/\epsilon^5)$ | SB |
| BGD-CE (This Work) | $\mathcal{O}(m^{2.8}T^{4/5})^\star$ | $\mathcal{O}(n^{13.5}m^{13}/\epsilon^5)$ | B |

All the aforementioned online learning algorithms concern general congestion games in which the strategy spaces of the agents do not admit any kind of combinatorial structure. As a result, *all of the above online learning algorithms require exponential time with respect to the number of resources.* For the important special case of Network Congestion Games over DAGs, there is a combinatorial structure that allows for polynomial time schemes as in (Awerbuch & Kleinberg, 2004; Fotakis et al., 2020; 2012; Angelidakis et al., 2013; Fotakis et al., 2015). We provide a variant of our algorithm that preserves the above guarantees while running in polynomial time with respect to the number of edges.

**Informal Theorem** *For Network Congestion games in Acyclic Directed Graphs (DAGs), Online Gradient Descent with Caratheodory Exploration, can be implemented in polynomial time.*

The above result follows from strategy spaces admitting polynomial size descriptions in this setting. We further exploit the specific structure of DAGs to compute exact 1-barycentric-spanners, which as noted in (Awerbuch & Kleinberg, 2004; Cesa-Bianchi & Lugosi, 2012) are not trivial to obtain for DAGs. We underline that exact spanners are not necessary, and the approximate method of (Awerbuch & Kleinberg, 2004) is perfectly suitable. However, our approach is simple, more efficient, and can be of independent interest.

**Our Techniques** The fundamental difficulty in designing no-regret online learning algorithms under bandit feedback is to guarantee that each resource is sufficiently explored. Unfortunately, standard bandit algorithms such as EXP3 (Auer et al., 2002) result in regret bounds of the form $\mathcal{O}(2^{m/2}\sqrt{T})$, that scale exponentially with respect to $m$. However, a long line of research in combinatorial bandits has produced algorithms with a regret polynomially dependent on $m$ (Awerbuch & Kleinberg, 2004; Dani et al., 2007a; György et al., 2007; Bubeck et al., 2012; Cesa-Bianchi & Lugosi, 2012; Kalai & Vempala, 2005; Neu & Bartók, 2013; Audibert et al., 2014). These algorithms, in order to overcome the exploration problem, use various techniques that can roughly be categorized two camps, simultaneous exploration ver-

sus alternating explore-exploit, as described by (Abernethy et al., 2009). However, to the best of our knowledge, none of these algorithms have been shown to converge to NE in congestion games once adopted by all agents.

Our online learning algorithm, guaranteeing both no-regret and convergence to equilibrium, is based on combining Online Gradient Descent (Zinkevich, 2003) with a novel exploration scheme, much like (Flaxman et al., 2004). Our exploration strategy is based on coupling the notion of barycentric spanners (Awerbuch & Kleinberg, 2004) with the notion of Bounded-Away Polytopes proposed by (Panageas et al., 2023) for the semi-bandit case. More precisely, (Panageas et al., 2023) introduced the notion of $\mu$-Bounded Away Polytope which corresponds to the description polytope of the strategy space (convex hull of all pure strategies) with the additional constraint that each resource is selected with probability at least $\mu > 0$. Projecting on this polytope ensures that the variance of the unobserved cost estimators remains bounded. In order to capture bandit estimators, we extend the notion of $\mu$-Bounded Away Polytope to denote the subset of the description polytope for which each point admits a decomposition with least $\mu$ weight on a preselected barycentric spanner $\mathcal{B}$.

This technique of projecting on $\mu$-Bounded polytopes closely ressembles the *mixing* strategies employed in online learning schemes that have alternating explore-exploit rounds. In those strategies, a fixed measure is added to bias the algorithm's chosen strategy. The projection on $\mu$-Bounded polytopes, however, scales the point before adding a bias, and, in some rounds, does not alter the point. It is therefore a mix of simultaneous and alternating exploration, depending on the round.

Finally, in order to provide a polynomial-time implementation of $OGD - CE$ for Network Congestion Games on Directed Acyclic Graphs we need exploit its well disposed combinatorial structure. In Section C.2, we propose a novel construction of barycentric spanners for DAGs that outputs a 1-barycentric spanner in polynomial time (see Algorithm 4) and yields an efficient selfish routing scheme that converges to an equilibrium.

## 2. Presentation of our formal result

In this section, we provide the necessary notation on congestion games and the bandit feedback model and to present the formal version of our result.

### 2.1. Congestion games

In congestion games, there exist a set of $n$ selfish agent and a set of $m$ resources $E$. Each agent $i \in [n]$ can select a subset of the resources $p_i \in S_i \subseteq 2^E$. A selection of resources $p_i \in S_i$ is also called a *pure strategy* while the set of all pure strategies $\mathcal{S}_i$ is also called *strategy space*. A selection of pure strategies profiles $p = (p_1, \ldots, p_n) \in \mathcal{S}_1 \times \cdots \times \mathcal{S}_n$ is called *joint strategy profile* and the set $\mathcal{S} := \mathcal{S}_1 \times \cdots \times \mathcal{S}_n$ is called *joint strategy space*. For a joint strategy profile $p \in \mathcal{S}$, we also use the notation $p = (p_i, p_{-i})$ to isolate (only in syntax) the strategy $p_i$ of agent $i$ from the rest of the strategies $p_{-i}$ of the other agents.

Given $p = (p_1, \ldots, p_n) \in \mathcal{S}$, the *load* of resource $e \in E$, denoted as $\ell_e(p)$, equals $\ell_e(p) = \sum_{i=1}^{n} \mathbb{1}\,(e \in p_i)$. and corresponds to the number of agents who have selected $e$ in their pure strategy. Each resource is additionally associated with a non-negative, non-decreasing *congestion cost function* $c_e : \mathbb{N} \to [0, c_{\max}]$ that associates a cost $c_e(\ell)$ for a given load $\ell$. For a joint strategy profile $p = (p_i, p_{-i}) \in \mathcal{S}$, the cost of agent $i \in [n]$ equals, $C_i(p_i, p_{-i}) = \sum_{e \in p_i} c_e(\ell_e(p_i, p_{-i}))$ and captures the congestion cost $c_e(\ell_e(p))$ of using resource $e \in p_i$.

**Definition 2.1** (Nash equilibrium)**.** A joint strategy profile $p = (p_1, \ldots, p_n) \in \mathcal{S}$ is called an $\epsilon$-approximate pure Nash equilibrium if and only if for all agents $i \in [n]$, $C_i(p_i, p_{-i}) \leq C_i(p'_i, p_{-i}) + \epsilon$ for any $p'_i \in \mathcal{S}_i$

To simplify notation we note that a pure strategy $p_i \in S_i$ can also be viewed as a $0/1$ vector $x^{p_i} \in \{0, 1\}^m$. Moreover given a joint strategy profile $p = (p_i, p_{-i}) \in \mathcal{S}_i$, we can construct a cost vector $c(\ell(p)) \in \mathbb{R}^m$ where $c_e(\ell(p)) = c_e(\ell_e(p_i, p_{-i}))$. Then the cost of agent $i \in [n]$ can be concisely described by an inner product as, $C_i(p_i, p_{-i}) = \sum_{e \in p_i} c_e(\ell_e(p_i, p_{-i})) = \langle c(\ell(p)), p_i \rangle$.. An agent $i \in [n]$ can also select a probability distribution over its pure strategies $\mathcal{S}_i$. Such a probability distribution $\pi_i \in \Delta(\mathcal{S}_i)$ is called a *mixed strategy*. Given joint mixed strategy profile $\pi = (\pi_i, \pi_{-i})$, the expected cost of agent $i$, equals $C_i(\pi_i, \pi_{-i}) := \mathbb{E}_{p \sim (\pi_i, \pi_{-i})}[C_i(p)]$. The notion of Nash Equilibrium provided in Definition 2.1 can be naturally extended in the context of mixed strategies.

**Definition 2.2** (Mixed Nash equilibrium)**.** A mixed joint strategy profile $\pi := (\pi_1, \ldots, \pi_n) \in \Delta(\mathcal{S}_1) \times \cdots \times \Delta(\mathcal{S}_n)$ is called an $\epsilon$-approximate mixed Nash equilibrium if and only if for all agents $i \in [n]$, $C_i(\pi_i, \pi_{-i}) \leq C_i(\pi'_i, \pi_{-i}) + \epsilon$ for any $\pi'_i \in \Delta(\mathcal{S}_i)$.

### 2.2. Bandit Dynamics in Congestion Games

When a congestion game is repeatedly played over $T$ rounds, each agent $i$ selects a new mixed strategy $\pi_i^t \in \Delta(\mathcal{S}_i)$ at each round $t \in [T]$ in their attempt to minimize their overall cost. The only feedback received by agent $i$ after picking $p_i^t$ is the cost $C_i(p_i^t, p_{-i}^t)$. This limited feedback is referred to as *bandit feedback* (Cui et al., 2022). This contrasts with the *full information feedback* where the agents observes the cost of *all* the available resources $\{c_e(\ell(p^t)) : \text{ for all } e \in E\}$ (Hazan, 2019) and the *semi-bandit feedback* setting where the agent observes the

cost of each of the individual resources it has selected $\{c_e(\ell(p^t)) : \text{for all } e \in p_i^t\}$ (Panageas et al., 2023).

Each agent $i \in [n]$ tries to selects the mixed strategies $\pi_i^t \in \Delta(\mathcal{S}_i)$ so as to minimize their overall cost over the $T$ rounds of play. Since the cost of the edges are determined by the strategies of the other agents that are unknown to agent $i$, the agent $i$ can assume that the cost of each agents are selected in an arbitrary and adversarial manner. Recalling that the cost $C_i(p_i^t, p_{-i}^t)$ is linear in $p_i^t$, the problem at hand is a particular instance of the Online Resource Selection under Bandit Feedback (Audibert & Bubeck, 2009).

The template of Online Resource Selection under Bandit Feedback is the following. Agent $i$ picks a mixed strategy $\pi_i^t \in \Delta(\mathcal{S}_i)$. An adversary picks a cost vector $c^t \in \mathbb{R}^m$, with $\|c^t\|_\infty \le c_{\max}$. Agent $i$ samples a pure strategy $p_i^t \sim \pi_i^t$ and incurs cost $l_i^t = \langle c^t, p_i^t \rangle$. Agent $i$ observes $l_i^t$ and updates its distribution $\pi_i^{t+1} \in \Delta(\mathcal{S}_i)$.

The agent's goal is therefore to output a sequence of strategies $p_i^{1:T}$ that minimize the incurred costs against *any* adversarially chosen sequence of cost vectors $c^{1:T}$ where $c^t$ can even depend on $\pi_i^{1:t-1}$. The quality of a sequence of play $p_i^{1:T}$ is measured in terms of *regret*, capturing its suboptimality with respect to the best fixed strategy.

**Definition 2.3** (Regret). The regret of the sequence $p_i^{1:T}$ with respect to the cost sequence $c^{1:T}$ equals $\mathcal{R}\left(p_i^{1:T}, c^{1:T}\right) := \sum_{t=1}^T \langle c^t, p_i^t \rangle - \min_{u \in \mathcal{S}_i} \sum_{t=1}^T \langle c^t, u \rangle$.

As already mentioned there are various online learning algorithms that even under the bandit feedback model are able guarantee sublinear regret almost surely. In the online learning literature such algorithms are called *no-regret*.

**Definition 2.4** (No-Regret). An online learning algorithm $\mathcal{A}$ for Linear Bandit Optimization is called no-regret if and only if for any cost vector sequence $c^1, \ldots, c^T$, $\mathcal{A}$ produces a sequence of mixed strategies $\pi_i^1, \ldots, \pi_i^T$ $(\pi_i^{t+1} = \mathcal{A}(l_i^1, \ldots, l_i^t))$ such that with high probability $\mathcal{R}\left(p_i^{1:T}, c^{1:T}\right) = o(T)$.

### 2.3. Our Results

The main contribution of our work is the design of a no-regret online learning algorithm under bandit feedback with the property that when adopted by all agents of a congestion game, leads to convergence to a Nash Equilibrium. The no-regret property of our algorithm is formally stated and established in Theorem 2.5 while its convergence properties to Nash Equilibrium are presented in Theorem 2.6.

**Theorem 2.5.** *There exists a no-regret algorithm, Bandit Gradient Descent with Caratheodory Exploration (BGD-CE) such that for any cost vector sequence $c_1, \ldots, c_T \in [0, c_{\max}]^m$ and $\delta > 0$, the regret $\mathcal{R}\left(p_i^{1:T}, c^{1:T}\right) :=$*

$\sum_{t=1}^T \sum_{e \in p_i^t} c_e^t - \min_{p_i^\star \in \mathcal{S}_i} \sum_{t=1}^T \sum_{e \in p_i^\star} c_e^t$ *verifies*

$$\mathcal{R}\left(p_i^{1:T}, c^{1:T}\right) \le \tilde{\mathcal{O}}\left(m^{5.5} c_{\max}^2 T^{4/5} \sqrt{\log \frac{1}{\delta}}\right)$$

*with probability $1 - \delta$.*

**Theorem 2.6** (Converge to NE). *Let $\pi^1, \ldots, \pi^T \in \Delta(S_1) \times \ldots \times \Delta(S_1)$ the sequence of strategy profiles produced if all agents adopt Bandit Gradient Descent with Caratheodory Exploration (BGD-CE). Then for all $T \ge \Theta\left(n^{13} m^{13}/\epsilon^5\right)$,*

$$\frac{1}{T}\mathbb{E}\left[\sum_{t=1}^T \max_{i \in [n]}\left[c_i(\pi_i^t, \pi_{-i}^t) - \min_{\pi_i \in \Delta(\mathcal{P}_i)} c_i(\pi_i, \pi_{-i}^t)\right]\right] \le \epsilon.$$

We note that the exact same notion of *best-iterate convergence* (as in Theorem 2.6) is considered in (Cui et al., 2022; Leonardos et al., 2022; Ding et al., 2022; Anagnostides et al., 2022c; Panageas et al., 2023). In Corollary 2.7 we present a clearer interpretation of Theorem 2.6.

**Corollary 2.7.** *In case all agents adopt BGD-CE for $T \ge \Theta(m^{13} m^{13}/\epsilon^5)$ then with probability $\ge 1 - \delta$,*

- *$(1 - \delta)T$ of the strategy profiles $\pi^1, \ldots, \pi^T$ are $\epsilon/\delta^2$-approximate Mixed NE.*

- *$\pi^t$ is an $\epsilon/\delta$-approximate Mixed NE once $t$ is sampled uniformly at random in $\{1, \ldots, T\}$*

The running time of $\text{BGD} - \text{CE}$ is exponential in general congestion games for which the strategy space $\mathcal{S}_i$ does not admit any combinatorial structure. In Theorem 2.8 we establish that for Network Congestion Games in Directed Acyclic Networks $\text{BGD} - \text{CE}$ can be implemented in polynomial time.

**Theorem 2.8.** *For Network Congestion Games over DAGs, $\text{BGD}-\text{CE}$ (Algorithm 3) can be implemented in polynomial time.*

The appendix is organized as follows. In Section B we present, BGD-CE (Algorithm 2) and explain the two main ideas behind its design. In Section C we present the polynomial-time implementation of BGD-CE (Algorithm 3) for the special case of Network Congestion Games over DAGs. Finally in Section D, we present the proofs for establishing Theorem 2.6 and Theorem 2.8.

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

## A. Related work

### A.1. Related Work

**Online Learning and Nash Equilibrium** Our work falls squarely within the recent line of research studying the convergence properties of online learning dynamics in the context of repeated games (Piliouras et al., 2022; Anagnostides et al., 2022a; Daskalakis et al., 2021; Anagnostides et al., 2022b; Farina et al., 2022; Hsieh et al., 2022; Zhou et al., 2018; Mertikopoulos & Zhou, 2019; Cohen et al., 2017). Specifically (Heliou et al., 2017; Palaiopanos et al., 2017; Mertikopoulos & Zhou, 2019; Zhou et al., 2018) establish asymptotic convergence guarantees for potential normal form games; congestion games are known to be isomorphic to potential games (Monderer & Shapley, 1996). Most of the aforementioned works use techniques from stochastic approximation and are orthogonal to ours. Furthermore, (Chen & Lu, 2016; Vu et al., 2021) study the convergence properties of first-order methods in non-atomic congestion games; non-atomic congestion games capture continuous populations and result in convex landscapes. On the other hand, atomic congestion games (the focus of this paper) result in non-convex landscapes.

**Bandits and Online Learning** As already mentioned, congestion games have been studied within the realm of online learning and bandits, where several no-regret algorithms have been proposed. The main difference between our and previous works is that, once the previously proposed algorithms are adopted by all agents, the overall system only converges to a Coarse Correlated Equilibrium and not necessarily to a Nash equilibrium as our algorithm guarantees (see (Panageas et al., 2023)). The design of no-regret algorithms for this setting began with (Awerbuch & Kleinberg, 2004) where a $O(T^{2/3})$ regret bound was achieved for linear bandit optimization against an oblivious adversary via introducing the notion of barycentric spanners. Follow up work (McMahan & Blum, 2004; György et al., 2007) built on this to propose a $O(T^{3/4})$ algorithms for linear bandits against *adaptive* adversaries. The optimal rates were then obtained by (Dani et al., 2007b) who establish $O(\sqrt{T})$ expected regret for the geometric hedge algorithm and closely followed by (Abernethy et al., 2009) who achieved the same expected regret using self-concordant barriers. Both these optimal rates were obtained with barriers (entropic or self-concordant) that diverge as points get close to the boundary of the strategy space. Unfortunately such barriers significantly degrade convergence rates to equilibria so we instead use $\ell_2$ regularization in our work.

Relatively recent papers have focused on providing *efficient* algorithms with *high-probability* guarantees against adaptive adversaries (Braun & Pokutta, 2016; Lee et al., 2020; Zimmert & Lattimore, 2022). See also (Cesa-Bianchi & Lugosi, 2012) for a general framework on combinatorial bandits.

**Existence and Equilibrium Efficiency** In the context of congestion games, the problem of equilibrium selection and efficiency has received a lot of interest. In (Koutsoupias & Papadimitriou, 1999), the notion of *Price of Anarchy* (PoA) was introduced that captures the ratio between the worst-case equilibrium and the optimal path assignment. Later works provided bounds on PoA (Roughgarden & Tardos, 2002; Christodoulou & Koutsoupias, 2005; Fotakis et al., 2005; de Keijzer et al., 2010; Bhawalkar et al., 2014; Mavronicolas & Spirakis, 2001) for both atomic and non-atomic settings. Another line of work has to do with the computational complexity of computing Nash equilibria in Network congestion games (Fabrikant et al., 2004; Ackermann et al., 2008; Klimm & Warode, 2020). Notably in (Fabrikant et al., 2004) it was shown that computing a Nash equilibrium in symmetric Network Congestion games can be done in polynomial time and also showed that in the asymmetric case, computing a pure Nash equilibrium belongs to class PLS (believed to be larger class than P). Further works appearred that investigate deterministic or randomized polynomial time approximation schemes for approximating a Nash equilibrium (Fotakis et al., 2009; 2008; Caragiannis et al., 2011; 2012; Caragiannis & Fanelli, 2019; Caragiannis & Jiang, 2023; Christodoulou et al., 2023; Giannakopoulos & Poças, 2023; Giannakopoulos et al., 2022; Kleer & Schäfer, 2021; Kleer, 2021; Audibert & Bubeck, 2009).

## B. Bandit Online Gradient Descent with Caratheodory Exploration

In this section, we present our online learning algorithm for general congestion games, called Bandit Online Gradient Descent with Caratheodory Exploration. The formal description of our algorithm lies in Section B.3 (Algorithm 2). We begin the section by introducing two essential ingredients. In Section B.1 we present the notion of Implicit Description Polytopes for Congestion Games and in Section B.2 the notion of Barycentric Spanners (Awerbuch & Kleinberg, 2004).

## B.1. Implicit Description and Strategy Sampling

The set of resources can be numbered such that $E = \{1, \ldots, m\}$. The latter allows for the strategy space $\mathcal{S}_i$ to be embedded in the vertices of the $m$ dimensional hypercube. Indeed any $p_i \in \mathcal{S}_i$ can be described, with a slight abuse of notation, by the vertex $p_i \in \{0, 1\}^m$ where $p_{i_e} = 1$ if and only if $e \in p_i$. The following definition formalizes this embedding.

**Definition B.1** (Implicit description polytope). For any element in $\mathcal{S}_i$, let $p_i \in \{0, 1\}^m$ denote its encoding as a vertex in the hypercube. The implicit description polytope $\mathcal{X}_i$ is given by the following convex hull

$$\mathcal{X}_i := \mathrm{conv}\left(\{p_i \in \{0, 1\}^m, \ p_i \in \mathcal{S}_i\}\right),$$

$\mathcal{X}_i$ is a closed convex polytope so there exists $A_i \in \mathbb{R}^{r_i \times m}$ and $d_i \in \mathbb{R}^{r_i}$, for some $r_i \in \mathbb{N}$, such that

$$\mathcal{X}_i = \{x \in \mathbb{R}^m, A_i x \le d_i\}$$

The polytope is therefore defined by the pair $(A_i, d_i)$ and its size is given by $r_i$ and $m$.

This implicit description polytope is of interest because the strategy space $\mathcal{S}_i$ corresponds to its extreme points. Moreover, the set of distribution over the strategy space $\Delta(\mathcal{S}_i)$ is also captured by the polytope as shown by the following definition.

**Definition B.2** (Marginalization). For any $\pi_i \in \Delta(\mathcal{S}_i)$ we can associate a point $x^{\pi_i} \in \mathcal{X}_i$ defined as

$$x^{\pi_i} = \sum_{p_i \in \mathcal{S}_i} \Pr_{u \sim \pi_i} [u = p_i] \, p_i.$$

The reverse correspondence of obtaining a distribution $\pi_i \in \Delta(\mathcal{S}_i)$ from a point $x_i \in \mathcal{X}_i$ can also established thanks to a result of Caratheodory (Carathéodory, 1907).

**Definition B.3** (Caratheodory decomposition). Let $x_i \in \mathcal{X}_i$. By Caratheodory's theorem, there exists $m + 1$ strategies $v_i^1, \ldots v_i^{m+1}$ and scalars $\lambda_1, \ldots, \lambda_{m+1}$ such that

$$x_i = \sum_{j=1}^{m+1} \lambda_j v_i^j \tag{CD}$$

with $\lambda_j \ge 0$ and $\sum_j \lambda_j = 1$. The set $\mathcal{C}_i = \left\{(v_i^1, \lambda_1), \ldots, (v_i^{m+1}, \lambda_{m+1})\right\}$ is called a Caratheodory decomposition of $x_i$

With the above, any point in $\mathcal{X}_i$ can be associated to a distribution that can be sampled easily.

## B.2. Barycentric Spanners and Bounded Away Polytopes

This section introduces the important concept of barycentric spanners (Awerbuch & Kleinberg, 2004). We will leverage barycentric spanners to ensure sufficient exploration of the resources set and hence guarantee low variance of the cost estimators.

**Definition B.4** ($\vartheta$-spanners). A subset of independent vectors $\{b_1, \ldots, b_s\} \subseteq \mathcal{X}_i$, with $s \le m$, is said to be $\vartheta$-spanner of $\mathcal{X}_i$, with $\vartheta \ge 1$, if, for all $x \in \mathcal{X}_i$, there exists $\alpha \in \mathbb{R}^s$ such that

$$x = \sum_{k=1}^{s} \alpha_k b_k \quad \text{and} \quad \alpha_i^2 \le \vartheta^2, \text{ for all } k \in [s].$$

Such collections of vectors can always be found as shown by the following theorem.

**Theorem B.5** (Existence of spanners ((Awerbuch & Kleinberg, 2004), Proposition 2.2)). *Any compact set $\mathcal{X}_i \subset \mathbb{R}^m$ admits an $O(1)$-spanner.*

We adopt barycentric spanners as a key ingredient in our algorithm. Since barycentric spanners essentially form a kind of basis of the polytope $\mathcal{X}_i$, we can introduce the basis polytope $\mathcal{D}_i$ in the following defintion.

**Definition B.6** (Basis polytope). Let $B_i$ be the matrix whose columns are $\vartheta$-barycentric spanners $b_1, \ldots, b_s$ of $\mathcal{X}_i$. The polytope defined as

$$\mathcal{D}_i = \{\alpha \in [-\vartheta, \vartheta]^s, \ B_i \alpha \in \mathcal{X}_i\}$$

is referred to as the basis polytope.

It is in this polytope that we can achieve fine control of norms necessary for our proofs, for this reason agents will operate in their respective basis polytopes. Moreover to ensure sufficient exploration, the boundaries of the polytope need to be avoided. More precisely, we introduce the notion of $\mu$-*Bounded-Away Basis Polytope* that will be central for our proposed algorithm.

**Definition B.7.** Let $\mu > 0$ be an exploration parameter. The $\mu$-*Bounded-Away basis Polytope*, denoted as $\mathcal{D}_i^\mu$, is defined as

$$\mathcal{D}_i^\mu \triangleq (1 - \mu)\mathcal{D}_i + \frac{\mu}{s}\mathbb{1}. \tag{1}$$

We remark that the $\mu$-Bounded-Away Polytope $\mathcal{D}_i^\mu$ is always non empty as it contains $\frac{1}{s}\mathbb{1}$, moreover, $\mathcal{D}_i^\mu \subseteq \mathcal{D}_i$. A simplified version of this idea has been shown successful for the semi-bandit feedback model (Panageas et al., 2023) and it appeared in (Chen et al., 2021) that used it in the context of online predictions with experts advice.

Equation (1) shows that any point $\alpha_i \in \mathcal{D}_i$ admits a decomposition where $\frac{1}{s}\mathbb{1}$ appears with coefficient $\mu$. Mapping back to the implicit description polytope, this implies that the point $x_i = B_i\alpha_i$ admits a decomposition that assigns a weight $\mu > 0$ to $\overline{b}_i = \frac{1}{|\mathcal{B}_i|}\sum_{b \in \mathcal{B}_i} b$, which can be understood as the uniform distribution over the spanners. In fact, there is a tractable way of obtaining this decomposition as evidenced by the following definition.

**Definition B.8** (Shifted Caratheodory decomposition). Given a barycentric spanner $\mathcal{B}_i$ and the respective $\mu$-bounded away basis polytope $\mathcal{D}_i$, for any $\alpha \in \mathcal{D}_i^\mu$, with $\alpha = (1 - \mu)z + \frac{\mu}{s}\mathbb{1}$ for some $z \in \mathcal{D}_i$, the shifted Caratheodory decomposition of $x = B_i\alpha$ is given by

$$x = (1 - \mu)\left[\sum_{(p,\lambda_p) \in \mathcal{C}_i} \lambda_p \cdot p\right] + \frac{\mu}{|\mathcal{B}_i|}\sum_{b \in \mathcal{B}_i} b_i$$

where $C_i$ is the Caratheodory decomposition of $B_i z \in \mathcal{X}_i$.

In Algorithm 1 we present how, for any $\alpha \in \mathcal{D}_i^\mu$, a point $x = B_i\alpha \in \mathcal{X}_i$ can be decomposed to a probability distribution $\pi_x \in \Delta(\mathcal{S}_i)$.

---
**Algorithm 1** `CaratheodoryDistribution`

---
**Input:** $x \in \mathcal{X}_i$, exploration parameter $\mu > 0$, spanner $\mathcal{B}_i = \{b_1, \ldots, b_s\}$. Consider the shifted decomposition of $x$ (see Definition B.8) with $\overline{b}_i = \frac{1}{|\mathcal{B}_i|}\sum_{b \in \mathcal{B}_i} b$, i.e.

$$x = (1 - \mu)\left(\sum_{(p,\lambda_p) \in \mathcal{C}_i} \lambda_p \cdot p\right) + \frac{\mu}{|\mathcal{B}_i|}\sum_{b \in \mathcal{B}_i} b_i$$

where $C_i = \{(\lambda_1, v_i^1), \ldots, (\lambda_{m+1}, v_i^{m+1})\}$ is the Caratheodory decomposition of $\frac{1}{1-\mu}(x - \frac{\mu}{|\mathcal{B}_i|}\sum_{b \in \mathcal{B}_i} b_i)$.
**Output** $\pi_x \in \Delta(\mathcal{S}_i)$ with $\text{supp}(\pi) = \{v_i^1, \ldots, v_i^{m+1}\} \cup \mathcal{B}_i$ such that

- $\Pr_{u \sim \pi_x}[u = v_k] = (1 - \mu)\lambda_k$ for all $k \in [m + 1]$

- $\Pr_{u \sim \pi_x}[u = b_s] = \frac{\mu}{|\mathcal{B}_i|}$ for all $b_s \in \mathcal{B}_i$

---

### B.3. Bandit Gradient Descent with Caratheodory Exploration

In this section we present our algorithm, called Bandit Gradient Descent with Caratheodory Exploration (BGD − CE) described in Algorithm 2.

Algorithm 2 and is based on Projected Online Gradient Descent (Zinkevich, 2003) but it includes two important variations leveraging the technical tools introduced in the previous sections.

*Resources sampling* In Step 6 of Algorithm 2 we need to sample from a distribution over $\mathcal{S}_i$. As this set can be exponentially large, this sampling procedure might have complexity exponential in $m$. To avoid such a computational complexity, we do

---

**Algorithm 2** Bandit Gradient Descent with Caratheodory Exploration and Bounded Away polytopes

---

Agent $i$ computes a $\mathcal{O}(1)$-barycentric spanner (see Definition B.4) $\mathcal{B} = \{b_1, \ldots, b_s\}$.

Agent $i$ sets $B_i \in \mathbb{R}^{m \times s}$ to be the matrix with columns $\{b_1, \ldots, b_s\}$.

Agent $i$ selects an arbitrary $\alpha_i^1 \in \mathcal{D}_i^{\mu_1}$.

**for** each round $t = 1, \ldots, T$ **do**

   Define $x_i^t = B_i \alpha_i^t$.

   Agent $i$ samples $p_i^t \sim \pi_i^t$ where $\pi_i^t = \texttt{CaratheodoryDistribution}(x_i^t; \mu_t, \mathcal{B})$ (Algorithm 1).

   Agent $i$ suffers cost, $l_i^t := \langle c^t, p_i^t \rangle$.

   Agent $i$ sets $\hat{c}^t \leftarrow l_i^t \cdot M_{i,t}^+ p_i^t$ where $M_{i,t} = \mathbb{E}_{v \sim \pi_i^t}[vv^\top]$.

   Agent $i$ updates $\alpha_i^{t+1} = \Pi_{\mathcal{D}_i^{\mu_{t+1}}}(\alpha_i^t - \gamma_t B_i^\top \hat{c}^t)$.

**end for**

---

not track distriutions but rather their maginalization $x_i^t$ and we sample from the Caratheodory distribution $\pi_i^t$ which has sparse support.

*Bounded variance estimator* Since we work under bandit feedback, we can not directly observe all the entries of the cost vector. To circumvent this challenge, we adopt the standard estimator for online linear optimization with bandit feedback proposed in (Dani et al., 2007b) which is $\hat{c}^t \leftarrow l_i^t \cdot M_{i,t}^+ p_i^t$ where $M_{i,t} = \mathbb{E}_{u \sim \pi_i^t}[uu^\top]$. The bounds on the variance of this estimator depends on the inverse of the smallest nonzero eigenvalue of $M_{i,t}$ (see Lemma E.1) but unfortunately this could be arbitrary small for points close to the boundaries of the polytope $\mathcal{X}_i$. For this reason, in Step 8 of Algorithm 2 we project on the set shrunk down polytope, $\mathcal{D}_i^\mu$, that ensures we are $\mu$ away from the boundary. Thanks to this, we can prove the following result concerning the cost estimator.

**Lemma B.9.** *The estimator $\hat{c}^t = l_i^t \cdot M_{i,t}^+ p_i^t$ satisfies*

1. $\mathbb{E}\left[\langle \hat{c}^t, x \rangle\right] = \langle c^t, x \rangle$ *for $x \in \mathcal{X}_i$*    *(Orthogonal Bias).*

2. $\|B_i^\top \hat{c}^t\|_2 \leq \vartheta \frac{m^{5/2}}{\mu_t} c_{\max}$.    *(Boundness).*

3. $\mathbb{E}\left[\|B_i^\top \hat{c}^t\|_2^2\right] \leq \frac{nm^4 c_{\max}^2}{\mu_t}$    *(Second Moment)*

Using Lemma B.9 we are able to establish both the no-regret property of Algorithm 2 as well as its convergence properties of Nash Equilibrium in case Algorithm 2 is adopted by all agents. In Theorem B.10 we formally stated and establish the no-regret property of Algorithm 2.

**Theorem B.10** (No-Regret). *Let $\delta \in (0,1)$. If agent $i \in [n]$ generates its strategies $p^{1:T}$ using $\texttt{Algorithm 2}$ with step sizes $\gamma_t = \sqrt{\frac{c_{\max}\mu_t}{\vartheta n^3 m^6 t}}$ and biases $\mu_t = \min\left\{\frac{n^{1/5}}{m^{7/5} t^{1/5} c_{\max}^{1/5}}, 0.5\right\}$, then, for any adversarial adaptive sequence $c^{1:T}$,*

$$\mathcal{R}\left(p_i^{1:T}, c^{1:T}\right) \leq \tilde{\mathcal{O}}\left(m^{5.5} c^2 T^{4/5} \sqrt{\log \frac{1}{\delta}}\right)$$

*with probability $1 - \delta$.*

In Theorem B.11 we establish the convergence properties of Algorithm 2 to Nash Equilibrium.

**Theorem B.11** (Convergence to Nash). *Let all the agents adopt $\texttt{Algorithm 2}$ with step sizes $\gamma_t = \sqrt{\frac{c_{\max}\mu_t}{n^3 m^6 t}}$ and biases $\mu_t = \frac{n^{1/5}}{m^{7/5} t^{1/5} c_{\max}^{1/5}}$. We denote by $\pi^1, \ldots, \pi^T$ the sequence of joint strategy profiles produced. Then, for $T \geq \Theta(m^{13} m^{13.5}/\epsilon^5)$,*

$$\frac{1}{T}\mathbb{E}\left[\sum_{t=1}^T \max_{i \in [n]}\left[c_i(\pi_i^t, \pi_{-i}^t) - \min_{\pi_i \in \Delta(\mathcal{P}_i)} c_i(\pi_i, \pi_{-i}^t)\right]\right] \leq \epsilon.$$

In Section D, we present the proof sketches of both Theorem B.10 and Theorem B.11.

We remark that the complexity of Algorithm 2 is polynomial with respect to the size of *implicit polytope* $\mathcal{X}_i$. However the for general congestion games the size of $\mathcal{X}_i$ can be exponential in $m$. Moreover constructing an $\mathcal{O}(1)$-barycentric spanner for general congestion games also requires exponential time in $m$ (Awerbuch & Kleinberg, 2004) when the size of the polytope is exponential. In the next section, we tailor the algorithm to cases when the polytope admits a convenient structure.

## C. Implementing Algorithm 2 in Polynomial-Time for DAGs

In this section we present how Algorithm 2 can be implemented in polynomial time for the special case of DAGs. The latter involves two key steps. The first one consists in computing barycentric spanners in polynomial time while the second in efficiently computing a Caratheorody Decomposition. We remark that none of the above steps can be done in polynomial time for general congestion games. To tackle the first challenge in Algorithm 4 we present a novel and efficient procedure for spanner construction which also consists the main technical contribution of this section. To tackle the second challenge, we use the approach introduced in the previous work of (Panageas et al., 2023). Overall, we present the computationally efficient version of Algorithm 2 for the case of Network Congestion Games over DAGs in Algorithm 3.

### C.1. Complexity for general congestion games

For $\vartheta = \mathcal{O}(1)$ but with $\vartheta > 1$, (Awerbuch & Kleinberg, 2004) shows that it is possible to compute a $\vartheta$-spanner for any compact set with a polynomial number of calls to a linear minimization oracle. The time complexity of this oracle depends polynomially on $r_i$ and $m$ where $r_i$ is the number of rows in $(A_i, d_i)$, the implicit description of $\mathcal{X}_i$. The updates of Algorithm 2 further require a Caratheodory decomposition for sampling at step 3, the inversion of a $m \times m$ matrix $M_{i,t}$ and finally a projection onto $\mathcal{D}_i$. Overall the complexity of a single update is therefore $\mathrm{poly}(r_i, m)$. For general congestion games, it can be the case that $r_i$ is exponential in $m$. For the special case of network games however, $\mathcal{X}_i$ corresponds to the flow polytope for which $r_i \leq m$. We discuss this special case in the next section.

### C.2. Efficient implementation of Algorithm 2 for DAGs

An efficient implementation is possible if the set of resources correspond to the edges of a DAG. First, recall that the implicit description polytope $\mathcal{X}_i$ admits a polynomial description. Indeed, in network congestion games $\mathcal{X}$ has the following simple form.

**Definition C.1** (Flow polytope)**.** The implicit description polytope of a Network Congestion Game over a *directed acyclic graph $G(V, E)$ with start and target node $s_i, t_i \in V$ is given by

$$\mathcal{X}_i \triangleq \left\{ x \in \{0,1\}^m : \sum_{e \in \mathrm{Out}(s_i)} x_e = 1 \right.$$
$$\sum_{e \in \mathrm{In}(v)} x_e = \sum_{e \in \mathrm{Out}(v)} x_e \quad \forall v \in V \setminus \{s_i, t_i\}$$
$$\left. \sum_{e \in \mathrm{In}(t_i)} x_e = 1 \right\}$$

Notice that the number of constraints is simply $|V|$. Therefore, a DAG admits an implicit description with $r_i = |V| < m$. Moreover, we have the following important characterization of the extreme points.

**Lemma C.2.** *(?)Lemma 11]panageas2023semi The extreme points of the $(s_i, t_i)$-path polytope $\mathcal{X}_i$ correspond to $(s_i, t_i)$-paths of $G(V, E)$ and vice versa.*

Therefore, despite the fact that there potentially exponentially many extreme points of $\mathcal{X}_i$, the set $\mathcal{X}_i$ is described concisely by $|V|$ constraints. The first important consequence of this result is that by invoking the following theorem we can ensure that Step 5 in Algorithm 2 runs in polynomial time.

**Theorem C.3.** *(Grötschel et al., 1988) Let $x \in \mathcal{X}_i = \{u \in [0,1]^m, \ A_i u \leq d_i\}$, with $A_i \in \mathbb{R}^{r_i \times m}$ and $d_i \in \mathbb{R}^m$. Then a Caratheodory decomposition can be computed in polynomial time with respect to $r_i$ and $m$.*

Given a shortest path algorithm, this can be done using (?)Algorithm 1]panageas2023semi. Moreover, also the projection in Step 8 of Algorithm 2 can be computed up to arbitrary accuracy in polynomial time given that $\mathcal{X}$ can be represented

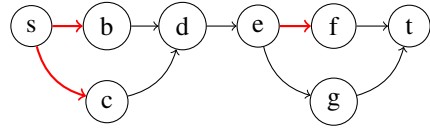

*Figure 1.* Construction of a 1-spanner for DAGs. We illustrate Algorithm 4 on a simple graph. We can select the three red edges as the, non-redundant, key edges. We cover these using 3 paths that will constitute the basis. For edge $s \to b$, we select $s \to b \to d \to e \to g \to t$. For the edge $s \to c$, we first check if is reachable from edge $s \to b$, we notice it is not. We then find a path starting from $s$. In this case, we select $s \to c \to d \to e \to g \to t$. For edge $e \to f$ we check if is reachable from the last covered edge (in topological order), we notice it is reachable from edge $s \to c$ so we select $s \to c \to d \to e \to f \to t$. The key idea we use to construct a 1-spanner is to ensure that when we cover edges, we first try to reach them with the previously covered edges going in reverse topological order. This prefix property ensures the 1-spanner property.

via $|V|$ affine constraints. The second computational bottleneck in the general case is the spanner computation. However, for the special case of DAGs, we present next an algorithm that construct exact 1-spanner which has better computational complexity compared to (Awerbuch & Kleinberg, 2004). The improvement is possible because the approach by (Awerbuch & Kleinberg, 2004) does not exploit the specific structure of DAGs although it is polynomial-time for DAGs. We propose, instead, an algorithm that stays in the natural parametrization of the problem and outputs a 1-spanner. The construction is detailed in Algorithm 4 and rests on a clever use of prefix paths. All in all, we have the next formal result.

**Theorem C.4.** *Given a Directed Acyclic Graph $G = (V, E)$ with source $s_i \in V$ and sink $t_i \in V$, there exists a polynomial time algorithm (i.e. Algorithm 4) computing an exact 1-spanner for $\mathcal{X}_i$.*

We give a constructive proof of Theorem C.4 in Section C.3. Overall, we propose the following simple algorithm that runs in polynomial time where the difference with the general case is that in Step 2 the spanner is computed efficiently by invoking Algorithm 4.

---

**Algorithm 3** Bandit Gradient Descent with Caratheodory Exploration and Bounded Away polytopes (Agent's $i$ perspective) **for DAGs**

---

**Input:** Step size sequence $(\gamma_t)_t$, bias coefficients $(\mu_t)_t$, a constant $\vartheta$.

Agent $i$ computes a 1-barycentric spanner $\mathcal{B} = \{b_1, \ldots, b_s\}$ with Algorithm 4.

Agent $i$ selects an arbitrary $x_i^1 \in \mathcal{X}_i$.

**for** each round $t = 1, \ldots, T$ **do**

    Agent $i$ sets $x_i^t = B_i \alpha_i^t$.

    Agent $i$ samples $p_i^t \sim \pi_i^t$ where $\pi_i^t = \texttt{CaratheodoryDistribution}(x_i^t; \mu_t, \mathcal{B})$ (Algorithm 1).

    Agent $i$ suffers cost, $l_i^t := \langle c^t, p_i^t \rangle$.

    Agent $i$ sets $\hat{c}^t \leftarrow l_i^t \cdot M_{i,t}^+ p_i^t$ where $M_{i,t} = \mathbb{E}_{v \sim \pi_i^t}[vv^\top]$.

    Agent $i$ updates $\alpha_i^{t+1}$ as, $\alpha_i^{t+1} = \Pi_{\mathcal{D}_i^{\mu_{t+1}}}(\alpha_i^t - \gamma_t B_i^T \hat{c}^t)$.

**end for**

---

## C.3. Constructing the spanner of DAGs

In this section we present Algorithm 4 that computes an 1-barycentric spanner for the special case of DAGs. To simplify notation for a given agent $i \in [n]$, we denote by $\mathcal{S}_i \subset \mathbb{R}^m$, the strategy space corresponding to set of all paths connecting $s_i$ to $t_i$. We can restrict our attention to the subgraph $G_i = (V_i, E_i)$ where $V_i$ and $E_i$ corresponds to the nodes and edges appearing in at least one path in $\mathcal{S}_i$.

### C.3.1. REDUNDANT EDGES

The convex hull of the strategy space $\mathcal{S}_i$ forms the path polytope $\mathcal{X}_i = \text{conv}(\mathcal{S}_i)$. This polytope is included in a subspace of $\mathbb{R}^m$ of dimension $m_i - n_i + 2$, where $n_i = |V_i|$. Indeed, for each node $v \in V \setminus \{s_i, t_i\}$, we can pick one outgoing edge $e_v^* \in \text{out}(v)$ such that for any $x \in \mathcal{P}_i$, we have

$$x_{e_v^*} = \sum_{e \in \text{in}(v)} x_e - \sum_{e \in \text{out}(v), e \neq e_v^*} x_e \tag{2}$$

for all $v \in V \backslash \{s_i, t_i\}$. These equations come from reasoning about flow preservation. Consequently, $\mathcal{X}_i$ belongs to the intersection of $n_i - 2$ hyperplanes, which is of dimension at most $m_i - n_i + 2$. In other words, although the strategy space is of dimension $m_i$, the degrees of freedom are restricted by the graph structure as some coordinates are redundant and predictable from other coordinates (see (2)). We single out these redundant edges in the following definition.

**Definition C.5.** For all $v \in V_i \backslash \{s_i, t_i\}$ (i.e all nodes except the source and termination nodes), we arbitrarily pick one edge denoted $e_v^* \in \text{out}(v)$ that will be referred to as a *redundant edge*.

The remaining edges will be referred to as a key edges. These key edges will aid us in constructing a 1-spanner. Indeed, from equation (2), we can see that the coordinates corresponding to redundant edges can be determined by the values at the key edges.

C.3.2. BASIS CONSTRUCTION

In order to construct the basis, we first need to perform a *topological ordering* of the nodes. A topological ordering of the nodes of a graph is a total ordering of the nodes such that for every directed edge with source vertex $u \in V$ and destination vertex $v \in V$, the node $u$ comes before $v$ in the ordering. We will use the $<$ symbol to denote such an ordering.

Let $v_1 = s_i, v_2, \ldots, v_n = t_i$ be a topological ordering of the nodes of $G_i$. This induces a topological ordering on the edges (sorted according to their origin node). We will construct a 1-spanner for $\mathcal{X}_i$ following this ordering. The following simple lemma (proved in Appendix H) about redundant paths will be essential.

**Definition C.6** (Redundant path). A path in $G_i$ is said to be a *redundant path* if consists entirely of redundant edges.

**Lemma C.7** (Redundant path lemma). *For any node $v_k \in V_i \backslash \{s_i\}$, there exists a redundant path connecting $v_k$ to $v_n = t_i$.*

We now have all the tools needed for the construction of the basis $b_1, \ldots, b_s$ where $s = m_i - n_i + 2$ is the total number of key edges. We provide the procedure in Algorithm 4.

---
**Algorithm 4** Edge covering basis

  **Input:** Key edges $e_1, \ldots, e_s$ in topological order.
  Basis $\leftarrow \varnothing$
  **for** $h = 1$ to $s$ **do**
    Let $p_{e_h \to t_i}$ be a *redundant path* connecting $\text{dest}(e_h)$ to $t_i$ (given by Lemma C.7).
    **for** $k = h - 1$ to $1$ **do**
      **if** there exists a path $p_{k \to h}$ joining $\text{dest}(e_k)$ to $\text{source}(e_h)$ **then**
        Set $b_h \leftarrow \text{Truncate}(b_k, e_k) \mid p_{k \to h} \mid p_{e_h \to t_i}$
        Set $\texttt{Prefix}(h) \leftarrow k$
        **break**
      **end if**
    **end for**
    **if** there is no preceding key edge connected to $e_h$ **then**
      Let $p_{s_i \to e_h}$ be a *redundant* path connecting $s_i$ to $\text{dest}(e_h)$.
      Set $b_h \leftarrow p_{s_i \to e_h} \mid p_{e_h \to t_i}$
      Set $\texttt{Prefix}(h) \leftarrow \bot$
    **end if**
    Basis $\leftarrow$ Basis $\cup \{b_h\}$
  **end for**
  **return** Basis

---

**Proposition C.8** (Prefix property). *Consider a covering basis generated by Algorithm 4. Let $e_k < e_l$ be two key edges. If $e_k$ and $e_l$ are connected in $G(V_i, E_i)$, then $\texttt{Prefix}(k) \neq \texttt{Prefix}(l)$ where $\texttt{Prefix}$ is the value set at lines 8 and 13 of Algorithm 4.*

This prefix property is the central ingredient needed to prove that the generated basis is a 1-barycentric spanner. Its proof can be found in Appendix H. With this, we can state the main result.

**Theorem C.9** (1-Spanner). *Let $b_1, \ldots, b_s$ be the covering basis generated by $\texttt{Algorithm}$ (4). For any $x \in \mathcal{X}_i$, there*

*exists $\alpha \in \mathbb{R}^s$ such that*

$$x = \sum_{h=1}^{s} \alpha_h b_i \qquad \text{and } \alpha_h^2 \leq 1$$

*Proof.* It suffices to prove the result for $x \in \mathcal{S}_i$, the extreme points of $\mathcal{X}_i$. Let $r_x = \text{Key}(x) \in \mathbb{R}^s$ where Key is the linear operator selecting the coordinates corresponding to the key edges. Correspondingly, let us define $r_1, \ldots, r_s$ such that

$$r_h = \text{Key}(b_h)$$

for $h = 1, \ldots, s$. Observe that the canonical basis vectors $v_1, \ldots, v_s$ of $\mathbb{R}^s$ can be expressed as

$$v_h = r_h - r_{\text{Prefix}(h)}$$

for $h = 1, \ldots, s$, and taking $r_\perp = 0_s$. Consequently,

$$r_x = \sum_{h \in r_x} v_h = \sum_{h \in r_x} \left( r_h - r_{\text{Prefix}(h)} \right) = \sum_{h=1}^{s} \alpha_h r_h$$

for some $\alpha \in \mathbb{R}^s$. Now it remains to prove that $|\alpha_h| \leq 1$. We know, by the prefix property C.8, that the mapping $\text{Prefix} : \{h : h \in r_x\} \to [s-1] \cup \{\perp\}$ is injective since the edges in $\{h : h \in r_x\}$ are connected. In other words, there are no duplicates in $\{\text{Prefix}(h), h \in r_x\}$. We express $r_x$ in the following convenient form.

$$r_x = \sum_{h \in r_x} r_h - \sum_{h \in \{\text{Prefix}(h), h \in r_x\}} r_h$$

With this, we can reason on a case by case basis for each coordinate as follows. Let $h \in [s]$. We first consider the case where $h \in r_x$. Since there are no duplicates, if we also have that $h \in \{\text{Prefix}(h), h \in r_x\}$, then $\alpha_h = 0$ otherwise $\alpha_h = 1$. Similarly, if $h \notin r_x$, then we either have $h \in \{\text{Prefix}(h), h \in r_x\}$ in which case $\alpha_h = -1$ or if not $\alpha_h = 0$. We thus find that $\alpha_h^2 \leq 1$. Now to conclude, we know from (2) that there exists a linear operator $\text{Fill} : \mathbb{R}^s \to \mathbb{R}^m$ that *fills* in the values of the redundant edges from the coordinate values of the key edges, hence $x = \text{Fill}(\text{Key}(x))$, which yields,

$$x = \text{Fill}\left[ \sum_{h=1}^{s} \alpha_h r_h \right] = \sum_{h=1}^{s} \alpha_h \text{Fill}[r_h] = \sum_{h=1}^{s} \alpha_h b_h.$$

$\square$

## D. Proof sketches

In this section we provide the basic steps for establishing Theorem B.10 and Theorem B.11.

### D.1. Regret analysis

The main observation needed to prove Theorem 1 is to notice that at Step 8 of Algorithm 2 the sequence $\alpha_i^{1:T}$ is obtained performing a close variant of Online Gradient Descent (OGD) on the sequence of gradient estimates $B^\top \hat{c}^{1:T}$. The subtle difference here is that the projection is done on $\mathcal{D}_i^{\mu_t}$, a time varying polytope. Luckily, a small variation in the analysis allows us to establish a guarantee similar to that of online gradient descent, with an added $\mu_t$ dependent error term.

We first slightly expand the definition of regret to include a fixed comparator $u \in \mathcal{X}_i$. We define the regret with respect to a comparator as follows

$$\mathcal{R}\left( p_i^{1:T}, c^{1:T}; u \right) := \sum_{t=1}^{T} \left\langle c^t, p_i^t - u \right\rangle.$$

It is easy to see that the regret defined earlier is obtained by taking the fixed action comparator $u^\star = \min_{u \in \mathcal{S}_i} \sum_{t=1}^{T} \left\langle c^t, u \right\rangle$, which is the best fixed action in hindsight. With this extended notion of regret, we can prove the following result on the approximate online gradient descent scheme performed by our algorithm.

**Lemma D.1** (Moving OGD). *Let $x_i^{1:T}$ and $\hat{c}_i^{1:T}$ be the sequences produced by* `Algorithm 2`,

$$\mathcal{R}\left(x_i^{1:T}, \hat{c}^{1:T}; u\right) \leq \frac{2m}{\gamma_T} + 2\sum_{t=1}^{T} \gamma_t \|\hat{c}^t\|_2^2 + 2mc_{\max}\sum_{t=1}^{T}\mu_t. \tag{3}$$

Now for us to use this result to control the regret of the algorithm, we have to pay attention to the following two points. First, the algorithm is not playing $x_i^{1:T}$ but rather the samples $p_i^{1:T}$ and, second, it is incurring costs with respect to $c^{1:T}$ and not $\hat{c}^{1:T}$. The regret of the algorithm is therefore measured by $\mathcal{R}\left(p_i^{1:T}, c^{1:T}; u\right)$. We have to relate this quantity to the regret bounded in equation (3). This can be done in two steps. The first is going from the samples $p_i^{1:T}$ to the marginalizations $x_i^{1:T}$.

**Lemma D.2** (First concentration lemma). *Let $p_i^1, \ldots, p_i^T \in \mathcal{P}_i$ be the sequences of strategies produced by* `Algorithm 2` *for the sequence of costs $c^1, \ldots, c^T$. We have with probability $1 - \delta$,*

$$\mathcal{R}\left(p_i^{1:T}, c^{1:T}; u\right) \leq \mathcal{R}\left(x_i^{1:T}, c^{1:T}; u\right) + c_{\max}m\sqrt{T\log\left(\frac{1}{\delta}\right)}. \tag{4}$$

All that remains now is swapping the cost vectors from the true $c^{1:T}$ to the estimated $\hat{c}^{1:T}$, which can be achieved by invoking a second concentration argument.

**Lemma D.3** (Second concentration lemma). *Let $\hat{c}^1, \ldots, \hat{c}^T$ the sequence produced in Step 7 of Algorithm 2 run on the sequence of costs $c^1, \ldots, c^T$. Then with probability $1 - \delta$,*

$$\mathcal{R}\left(x_i^{1:T}, c^{1:T}; u\right) \leq \mathcal{R}\left(x_i^{1:T}, \hat{c}^{1:T}; u\right) + m^3 c_{\max}\vartheta^{3/2}\sqrt{\sum_{t=1}^{T}\frac{1}{\mu_t^2}\log(1/\delta)}. \tag{5}$$

Now to prove Theorem B.10, it suffices to simply plug (5) inside (4) to upper bound the regret of the algorithm with the regret of online gradient descent. Then, invoking Lemma D.1 which controls the regret of the latter, we can obtain bound on the regret of the algorithm with respect to a comparator $u \in \mathcal{X}_i$. To conclude and obtain B.10, a simple union bound over all $u \in \mathcal{X}_i$ yields the result. We detail the proof in Appendix F.

### D.2. Convergence to Nash (Proof of Theorem B.11)

In this section, we prove Theorem B.11. We will be using the fact that congestion games always admit a *potential function* (Monderer & Shapley, 1996) capturing the change in cost when a sole agent alters its strategy. The potential function of congestion games is given by the following function.

**Theorem D.4.** *The potential function $\Phi : \mathcal{S} \to \mathbb{R}_+$ given by $\Phi(p) = \sum_e \sum_{i=1}^{\ell_e(p)} c_e(i)$, has the property that $C_i(p_i', p_{-i}) - C_i(p_i, p_{-i}) = \Phi(p_i', p_{-i}) - \Phi(p_i, p_{-i})$.*

The key observation here is that the potential function is a *shared* function that measures the change in cost when any agent deviates from a joint profile. This same function also captures the change in *expected* cost once it is viewed as a function over the polytope $\mathcal{X} \triangleq \mathcal{X}_1 \times \cdots \times \mathcal{X}_n$.

**Definition D.5.** *The function $\Phi : \mathcal{X} \to \mathbb{R}_+$, defined as $\Phi(x) = \sum_{\mathcal{S}\subseteq[n]}\prod_{j\in\mathcal{S}}x_{je}\prod_{j\notin\mathcal{S}}(1 - x_{je})\sum_{\ell=0}^{|\mathcal{S}|}c_e(\ell)$ verifies*

$$C_i(\pi_i, \pi_{-i}) - C_i(\pi_i', \pi_{-i}) = \Phi(x_i, x_{-i}) - \Phi(x_i', x_{-i})$$

*for any $\pi \in \Delta(\mathcal{S}_1) \times \cdots \times \Delta(\mathcal{S}_n)$, with marginilization $x \in \mathcal{X}$, and any $i \in [n]$, where $\pi_i' \in \Delta(\mathcal{S}_i)$, with marginalization $x_i'$.*

The function $\Phi$ is not convex over $\mathcal{X}$ but it is smooth making it friendly to gradient based optimization. We can show that the function $\Phi$ is differentiable and its gradient $\nabla\Phi$ is Lipschitz continuous with constant $(2n^2\sqrt{m}c_{\max})$. However, since we operate in the basis polytope $\mathcal{D} = \mathcal{D}_1 \times \cdots \times \mathcal{D}_n$, we are interested in the function $\tilde{\Phi}$ defined as

$$\tilde{\Phi} : \alpha \mapsto \Phi(B\alpha),$$

where $B$ is the block diagonal matrix with $B_1, \ldots, B_n$ as its diagonal elements. This function inherits all the nice properties of $\Phi$ up to some additional factors. Indeed with a simple computation, we can show the following result.

**Proposition D.6.** *The function $\tilde{\Phi}$ is $\frac{1}{\lambda}$-smooth with $\lambda = (2n^2 m^{7/2} c_{\max})^{-1}$.*

Stationary points of $\Phi$ correspond to Nash equilibria (Monderer & Shapley, 1996), thus making the function $\Phi$ the essential tool used for proving our result. Indeed in the sequel we technically prove convergence to stationary points of the potential function. Stationary points are defined as follows.

**Definition D.7** (Stationarity). *A point $\alpha \in \mathcal{D}^\mu$ is called an $(\epsilon, \mu)$-stationary point if*

$$G^\mu(\alpha) \triangleq \left\| \alpha - \Pi_{\mathcal{D}^\mu} \left[ \alpha - \frac{\lambda}{2} \nabla \tilde{\Phi}(\alpha) \right] \right\|_2 \le \epsilon.$$

Given an $(\epsilon, \mu)$-stationary point $\alpha$, then any mixed strategy with marginalization $x = B\alpha$ is an approximate mixed Nash equilibrium. We formalize this in the following result.

**Proposition D.8** (From Stationarity to Nash). *Let $\pi \in \Delta(\mathcal{S}_1) \times \cdots \times \Delta(\mathcal{S}_n)$. Let $x \in \mathcal{X}$ be the marginalization of $\pi$. If $x = B\alpha$, with $\alpha \in \mathcal{D}$ an $(\epsilon, \mu)$-stationary point, then $\pi$ is a $4n^{2.5} m^4 c_{\max} (\epsilon + \mu)$-mixed Nash equilibrium.*

We have thus reduced the problem of finding mixed nash equilibria to that of finding stationary points of $\tilde{\Phi}$. We will find such stationary points by studying the joint vector of the iterates. We initiate our study by recalling the notation of the joint strategies of the players. For each $t \in [T]$, we collect each player's iterates in one vector in $\mathcal{D}$ defined as $\alpha^t \triangleq [\alpha_1^t, \ldots, \alpha_n^t]$. It is easy to see that when all players play according to Algorithm 2, the produced sequence of vectors $\alpha^1, \ldots, \alpha^T$ verifies

$$\alpha^{t+1} = \Pi_{\mathcal{D}^{\mu_{t+1}}} \left[ \alpha^t - \gamma_t \cdot \nabla_t \right] \tag{6}$$

where $\nabla_t \triangleq \left[ B_1^\top \hat{c}_1^t, \ldots, B_n^\top \hat{c}_n^t \right]$. It turns out that that $\nabla_t$ is an estimator for $\nabla \tilde{\Phi}$ as shown by the following lemma.

**Lemma D.9** (Estimator property). *Let $t \in [T]$ and $\mathcal{F}_t$ be the sigma-field generated by $\alpha_1, \ldots, \alpha_t$ and denote the conditional expectation as $\mathbb{E}_t[\cdot] \triangleq \mathbb{E}[\cdot | \mathcal{F}_t]$. It holds that*

*1. $\mathbb{E}_t[\nabla_t] = \nabla \tilde{\Phi}(\alpha^t)$,*

*2. $\mathbb{E}_t[\|\nabla_t\|_2^2] \le \frac{nm^4 c_{\max}^2}{\mu_t}$*

Our goal will be to show that the sequence $\alpha^1, \ldots, \alpha^T$ visits a point with a small stationarity gap. To prove this, the time varying Moreau envelope $M_{\lambda \tilde{\Phi}}^t$ of $\tilde{\Phi}$, defined as

$$M_{\lambda \tilde{\Phi}}^t(\alpha) \triangleq \min_{y \in \mathcal{D}^{\mu_t}} \left\{ \tilde{\Phi}(y) + \frac{1}{\lambda} \|\alpha - y\|_2^2 \right\},$$

will play a central role as is shown by the following lemma.

**Lemma D.10** (Gap control). *Let $G^t(\alpha) := \|\Pi_{\mathcal{D}^{\mu_t}} \left[ \alpha - \frac{\lambda}{2} \nabla \tilde{\Phi}(\alpha) \right] - x\|_2$ denote the $\mu_t$-stationarity gap. We have that for any $\alpha \in \mathcal{D}^{\mu_t}$,*
$$G^t(\alpha) \le \lambda \|\nabla M_{\lambda \tilde{\Phi}}^t(\alpha)\|_2$$

Controlling the stationarity gap of an iterate therefore boils down to bounding the norm of the gradient of $M_{\lambda \tilde{\Phi}}^t$ along the sequence. By observing that the update rule (6) closely corresponds to performing stochastic gradient descent step on $M_{\lambda \tilde{\Phi}}^t$, we are able to show the following result.

**Theorem D.11** (Stochastic gradient descent). *Consider the sequence $\alpha^1, \ldots, \alpha^T$ produced by Equation 6. Then,*

$$\frac{1}{T} \sum_{t=1}^T \mathbb{E} \left[ \|\nabla M_{\lambda \tilde{\Phi}}^t(\alpha^t)\|_2 \right] \le 2n^{1.5} \sqrt{\frac{2m^{1.5} c_{\max}}{\gamma_T T} + \frac{n^3 m^{7.5}}{\gamma_T T} \sum_{t=1}^T \frac{\gamma_t^2}{\mu_t}}$$

Finally, in order to obtain Theorem B.11, it suffices to combine the stochastic gradient descent result in Theorem D.11 with Lemma D.10 and observe that the sequence of iterates visits a point with a small stationarity gap. Combining this with proposition D.8 which relates stationarity to Nash equilibria yields the result. We provide a complete proof in section G.2.

## E. Properties of the estimator $\hat{c}^t$

The central difficulty of *bandit* feedback lies in the construction of a low variance estimator for the unobserved cost vector $c^t$ at each round $t \in [T]$. In what follows we prove two results on $\hat{c}^t$, the estimator constructed in step 7 of Algorithm 2 that will be instrumental to both the regret analysis and the convergence to equilibrium.

First we show that the estimator is bounded almost surely.

**Lemma E.1** (Bounded estimator). *For any $t \in [T]$, the estimator $\hat{c}^t = l_i^t \cdot M_{i,t}^+ p_i^t$ is almost surely bounded and*

$$\|B_i^\top \hat{c}^t\|_2 \le \vartheta \frac{m^{5/2}}{\mu_t} c_{\max}.$$

*Proof.* Let $i \in [n], t \in [T]$. Recall that $B_i \in \mathbb{R}^{m \times s}$ is the matrix whose columns are the $s$ elements of the barycentric spanner. Let us write $M_{i,t}$ in a more convenient form. Recall that $\pi_i^t$ is the Caratheodory distribution computed by Algorithm 1. It then follows (from step 3 in Algorithm 1) that

$$\pi_i^t = (1 - \mu_t)\tau_i^t + \mu_t \nu_i$$

where $\nu_i$ is the uniform distribution over the barycentric spanners and $\tau_i$ is the distribution supported on the Caratheodory decomposition. We can then express $M_{i,t}$ as follows.

$$\begin{aligned}
M_{i,t} &= \mathbb{E}_{u \sim \pi_i^t}\left[uu^\top\right] \\
&= (1 - \mu_t)\mathbb{E}_{u \sim \tau_i^t}\left[uu^\top\right] + \mu_t \mathbb{E}_{u \sim \nu_i}\left[uu^\top\right] \\
&= (1 - \mu_t)B_i\left(\mathbb{E}_{u \sim \tau_i^t}\left[\alpha_u \alpha_u^\top\right]\right)B_i^\top + \frac{\mu_t}{s}B_i\left(\sum_{k=1}^s e_k e_k^\top\right)B_i^\top \\
&= B_i N_{i,t} B_i^\top
\end{aligned}$$

where we defined $N_{i,t} := (1 - \mu_t)\mathbb{E}_{u \sim \tau_i^t}\left[\alpha_u \alpha_u^\top\right] + \frac{\mu_t}{s}I_s$. Notice here that it is easy to see that $N_{i,t} \succeq \frac{\mu_t}{s}I_s$ which implies that

$$N_{i,t}^+ \preceq \frac{s}{\mu_t}I_s. \tag{7}$$

Now, since $B_i$ has independent columns, we have that

$$M_{i,t}^+ = \left(B^\top\right)^+ N_{i,t}^+ B^+ \tag{8}$$

Moreover, we know there exists $\alpha_{i,t} \in \mathbb{R}^s$ such that $p_i^t = B\alpha_{i,t}$. With these in hand, let us analyze the estimator $\hat{c}^t$. We have that

$$\hat{c}^t = \langle c^t, p_i^t\rangle M_{i,t}^+ p_i^t = \langle c^t, p_i^t\rangle M_{i,t}^+ B\alpha_{i,t}$$

By plugging in (8), we find that

$$B_i^\top \hat{c}^t = \langle c^t, p_i^t\rangle N_{i,t}^+ \alpha_{i,t} \tag{9}$$

Consequently,

$$\left\|B_i^\top \hat{c}^t\right\| \le mc_{\max}\vartheta\frac{s^{3/2}}{\mu_t}$$

which allows us to conclude by using that using $s \le m$. $\qquad\square$

**Lemma E.2** (Orthogonal Bias). *For any $t \in [T]$, for any $x \in \mathcal{X}_i$,*

$$\langle c^t - \mathbb{E}_{\pi_i^t}[\hat{c}^t], x\rangle = 0.$$

*Proof.* Let $M = \mathbb{E}_{\pi_i^t}\left[pp^\top\right]$. Recall that $\hat{c}^t = M^+ p_i^t \langle p_i^t, c^t\rangle$. We have that

$$\mathbb{E}_{\pi_i^t}[\hat{c}^t] = M_{i,t}^+ M_{i,t} c^t = \left(B_i^\top\right)^+ B_i^\top c^t.$$

where the second equality is obtained using (8). It follows that for any $x \in \mathcal{X}_i$, which we know can be written $x = B_i \alpha_x$, we have that

$$\left\langle M_{i,t}^+ M_{i,t} c^t, x \right\rangle = \left\langle \left( B_i^\top \right)^+ B_i^\top c^t, x \right\rangle = \left\langle c, B_i B_i^+ x \right\rangle$$
$$= \left\langle c, B_i B_i^+ B_i \alpha_x \right\rangle \quad = \left\langle c, x \right\rangle$$

where the last line follows from the fact that $B_i^+$ is a right inverse when $B_i$ has independent columns, which is true by construction. □

## F. Regret analysis: Proof of Theorem B.10

In this section, we provide a complete proof of the regret bound. We first prove the two lemmas that relate the regret of the algorithm to the quantity bounded by the moving online gradient descent lemma. We then prove the online gradient descent lemma and conclude the section with a complete proof of Theorem B.10.

**Lemma F.1** (First concentration lemma). *Let $p_i^1, \ldots, p_i^T \in \mathcal{P}_i$ be the sequences of strategies produced by* `Algorithm 2` *for the sequence of costs $c^1, \ldots, c^T$. We have with probability $1 - \delta$,*

$$\mathcal{R}\left( p_i^{1:T}, c^{1:T}; u \right) \leq \mathcal{R}\left( x_i^{1:T}, c^{1:T}; u \right) + c_{\max} m \sqrt{T \log\left( \frac{1}{\delta} \right)}. \tag{4}$$

*Proof.* The result is obtained by a straightforward application of Azuma-Hoeffding's inequality. Indeed,

$$\mathbb{E}_t \left[ \left\langle c^t, p_i^t \right\rangle - \left\langle c^t, x_i^t \right\rangle \right] = 0$$

and $|\left\langle c^t, p_i^t \right\rangle - \left\langle c^t, x_i^t \right\rangle| \leq m c_{\max}$ almost surely. The sequence $(\left\langle c^t, p_i^t \right\rangle - \left\langle c^t, x_i^t \right\rangle)_t$ is a sequence of bounded martingale increments. We can thus apply Azuma-Hoeffding's inequality. □

The following second lemma swaps out the real cost vectors with their estimates.

**Lemma F.2** (Second concentration lemma). *Let $\hat{c}^1, \ldots, \hat{c}^T$ the sequence produced in Step 7 of Algorithm 2 run on the sequence of costs $c^1, \ldots, c^T$. Then with probability $1 - \delta$,*

$$\mathcal{R}\left( x_i^{1:T}, c^{1:T}; u \right) \leq \mathcal{R}\left( x_i^{1:T}, \hat{c}^{1:T}; u \right) + m^3 c_{\max} \vartheta^{3/2} \sqrt{\sum_{t=1}^{T} \frac{1}{\mu_t^2} \log(1/\delta)}. \tag{5}$$

*Proof.* This result is again a straightforward application of Azuma-Hoeffding's concentration inequality. Indeed, by the *Orthogonal Bias Lemma* E.2, we have that

$$\mathbb{E}_t \left[ \left\langle c^t - \hat{c}^t, x_i^t - u \right\rangle \right] = 0$$

It remains to show that $|\left\langle c^t - \hat{c}^t, x_i^t - u \right\rangle|$ is bounded almost surely. Since $B_i$ is a $\vartheta$-spanner, notice that there exists $\alpha^u \in \mathbb{R}^s$ such that $u = B\alpha^u$. We can thus write

$$|\left\langle c^t - \hat{c}^t, x_i^t - u \right\rangle| = |\left\langle B_i^\top \left( c^t - \hat{c}^t \right), \alpha_i^t - \alpha^u \right\rangle|$$
$$\leq \| B_i^\top \left( c^t - \hat{c}^t \right) \|_2 \| \alpha_i^t - \alpha^u \|_2,$$

where the last inequality was obtained by Cauchy-Schwartz. Now recalling the definition of $\hat{c}^t$, we have that

$$B_i^\top \left( c^t - \hat{c}^t \right) = \left( B_i^\top - B_i^\top M_{i,t}^+ B_i \alpha_{i,t} \alpha_{i,t}^\top B_i^\top \right) c^t$$
$$= \left( I - B_i^\top M_{i,t}^+ B_i \alpha_{i,t} \alpha_{i,t}^\top \right) B_i^\top c^t$$

Recalling (8), we have that

$$\left( I - B_i^\top M_{i,t}^+ B_i \alpha_{i,t} \alpha_{i,t}^\top \right) \preceq |1 - \vartheta^2 \frac{s^2}{\mu_t}| I_m \preceq \vartheta^2 \frac{s^2}{\mu_t} I_m$$

for $\mu_t \leq s^2 \vartheta$. We therefore get that

$$\| B_i^\top \left( c^t - \hat{c}^t \right) \|_2 \leq \vartheta^2 \frac{s^{5/2} c_{\max}}{\mu_t}$$

This allows us to conclude that

$$\left\langle c^t - \hat{c}^t, x_i^t - u \right\rangle \leq \frac{m^3 c_{\max} \vartheta^3}{\mu_t}$$

(using $s \leq m$). The sequence $(\langle c^t - \hat{c}^t, x_i^t - u \rangle)_t$ is therefore a bounded sequence of martingale increments. We can apply Azuma-Hoeffding's inequality. $\qquad\square$

By plugging (5) into (4), we have reduced the problem of bounding the regret to controlling the regret of moving OGD given by $\mathcal{R}\left(x_i^{1:T}, \hat{c}^{1:T}; u\right)$.

**Lemma F.3** (Moving OGD). *Let $x_i^{1:T}$ and $\hat{c}_i^{1:T}$ be the sequences produced by* `Algorithm 2`,

$$\mathcal{R}\left(x_i^{1:T}, \hat{c}^{1:T}; u\right) \leq \frac{2m}{\gamma_T} + 2\sum_{t=1}^{T} \gamma_t \|\hat{c}^t\|_2^2 + 2m c_{\max} \sum_{t=1}^{T} \mu_t. \tag{3}$$

*Proof.* The idea here will be to relate $\alpha_i^{1:T}$ to a sequence that is almost performing Online Gradient Descent on the fixed polytope $\mathcal{D}_i$. To this end, we introduce the auxiliary sequence $\tilde{\alpha}_i^{1:T}$ defined as

$$\tilde{\alpha}_i^t = \frac{1}{1 - \mu_t}(\alpha_i^t - \frac{\mu_t}{s}\mathbb{1})$$

and its corresponding point $\tilde{x}_i^t = B_i \tilde{\alpha}_i^t$. Since $\alpha_i^t \in \mathcal{D}_i^{\mu_t}$, we have that $\tilde{\alpha}_i^t \in \mathcal{D}_i$. Moreover, a simple re-arrangement gives $\alpha_i^t = (1 - \mu_t)\tilde{\alpha}_i^t + \frac{\mu_t}{s}\mathbb{1}$ With this in hand, we can write that

$$\left\langle \hat{c}^t, x_i^t - u \right\rangle = (1 - \mu_t)\left\langle \hat{c}^t, \tilde{x}_i^t - u \right\rangle + \mu_t \left\langle \hat{c}^t, \bar{b}_i \right\rangle$$
$$\leq \left\langle (1 - \mu_t)\hat{c}^t, \tilde{x}_i^t - u \right\rangle + m c_{\max} \mu_t$$
$$\leq \left\langle \hat{c}^t, \tilde{x}_i^t - u \right\rangle + 2m c_{\max} \mu_t$$

It then follows that

$$\mathcal{R}\left(x_i^{1:T}, \hat{c}^{1:T}; u\right) \leq \mathcal{R}\left(\tilde{x}_i^{1:T}, \hat{c}^{1:T}; u\right) + 2m c_{\max} \sum_{t=1}^{T} \mu_t \tag{10}$$

It remains to show that this regret term of the auxiliary sequence is controllable. This will follow from a simple observation on the update rule. Recall that this update rule in `Step 8` of Algorithm 2 is given by

$$\alpha_i^{t+1} = \Pi_{\mathcal{D}^{\mu_{t+1}}}\left[\alpha_i^t - \gamma_t B_i^\top \hat{c}^t\right]$$

By Lemma I.1, we know that we can express $\Pi_{\mathcal{D}_i^{\mu_{t+1}}}$ in terms of $\Pi_{\mathcal{D}_i}$, which allows us to write that

$$\alpha_i^{t+1} = (1 - \mu_{t+1})\Pi_{\mathcal{D}_i}\left[\frac{1}{1 - \mu_{t+1}}(\alpha_i^t - \gamma_t B_i^\top \hat{c}^t - \frac{\mu_t}{s}\mathbb{1})\right] + \frac{\mu_t}{s}\mathbb{1}$$

Rearranging we find that

$$\tilde{\alpha}_i^{t+1} = \Pi_{\mathcal{D}_i}\left[\tilde{\alpha}_i^t - \frac{\gamma_t}{1 - \mu_{t+1}}B_i^\top \hat{c}^t + (\mu_{t+1} - \mu_t)\left(\frac{\alpha_i^t - \frac{1}{s}\mathbb{1}}{(1 - \mu_t)(1 - \mu_{t+1})}\right)\right]$$

The last term in the projection is an error term that can easily be handled, we denote it by $e_t := \left(\frac{\alpha_i^t - \frac{1}{s}\mathbb{1}}{(1 - \mu_t)(1 - \mu_{t+1})}\right)$. We thus have that the auxiliary sequence is performing online gradient descent with a small error term since

$$\tilde{\alpha}_i^{t+1} = \Pi_{\mathcal{X}}\left[\tilde{\alpha}_i^t - \tilde{\gamma}_t B_i^\top \hat{c}^t + (\mu_{t+1} - \mu_t)e_t\right]$$

where $\tilde{\gamma}_t := \frac{\gamma_t}{1 - \mu_{t+1}}$. To control the regret of this approximate OGD, we consider the regret incurred on a single update.

Recall that $u \in \mathcal{X}_i$ and that there exists $\alpha^u \in \mathcal{D}_i$ such that $u = B_i \alpha^u$. We know by the contractive property of the projection that

$$\|\tilde{\alpha}_i^{t+1} - \alpha^u\|_2^2 \leq \|\tilde{\alpha}_i^t - \alpha^u - \tilde{\gamma}_t B_i^\top \hat{c}^t + (\mu_{t+1} - \mu_t)e_t\|_2^2$$
$$\leq \|\tilde{\alpha}_i^t - \alpha^u\|_2^2 - 2\tilde{\gamma}_t \left\langle \hat{c}^t, \tilde{x}_i^t - u \right\rangle + 2\tilde{\gamma}_t^2 \|B_i^\top \hat{c}^t\|_2^2 + 2(\mu_{t+1} - \mu_t)\left\langle e_t, \tilde{\alpha}_i^t - \alpha^u \right\rangle + 2(\mu_{t+1} - \mu_t)^2 \|e_t\|_2^2$$

where the second inequality follows from Young's inequality. Now since $0 \leq \mu_t \leq \frac{1}{2}$ for $t \geq \frac{32m^4 n}{c_{\max}}$, we have that $\|e_t\|_2 \leq 2\sqrt{m}$ and $(\mu_{t+1} - \mu_t)^2 \leq \frac{1}{2}(\mu_t - \mu_{t+1})$. Consequently,

$$\|\tilde{\alpha}_i^{t+1} - \alpha^u\|_2^2 \leq \|\tilde{\alpha}_i^t - \alpha^u\|_2^2 - 2\tilde{\gamma}_t \langle \hat{c}^t, \tilde{x}_i^t - u \rangle + 2\tilde{\gamma}_t^2 \|B_i^\top \hat{c}^t\|_2^2 + 8m(\mu_t - \mu_{t+1})$$

Rearranging, we obtain that

$$\langle \hat{c}^t, \tilde{x}_i^t - u \rangle \leq \frac{1}{2\tilde{\gamma}_t} \left( \|\tilde{\alpha}_i^t - \alpha^u\|_2^2 - \|\tilde{\alpha}_i^{t+1} - \alpha^u\|_2^2 \right) + \tilde{\gamma}_t \|B_i^\top \hat{c}^t\|_2^2 + \frac{8m}{\tilde{\gamma}_t}(\mu_t - \mu_{t+1})$$

By summing from $t = \bar{t} := \frac{32m^4 n}{c_{\max}}$ to $t = T$ and using the telescoping Lemma I.3, we find that

$$\mathcal{R}\left(\tilde{x}_i^{\bar{t}:T}, \hat{c}^{\bar{t}:T}; u\right) \leq \frac{5m}{\gamma_T} + 2\sum_{t=\bar{t}}^{T} \gamma_t \|B_i^\top \hat{c}^t\|_2^2$$

where we have used the fact that $\gamma_t \leq \tilde{\gamma}_t \leq 2\gamma_t$ and $m \geq 2$ to simplify the expression. Finally, using that

$$\mathcal{R}\left(\tilde{x}_i^{1:\bar{t}}, \hat{c}^{1:\bar{t}}; u\right) \leq 32nm^4,$$

we conclude that

$$\mathcal{R}\left(\tilde{x}_i^{1:T}, \hat{c}^{1:T}; u\right) \leq \frac{5m}{\gamma_T} + 2\sum_{t=1}^{T} \gamma_t \|\hat{c}\|_2^2 + 32nm^4$$

We obtain the result by plugging the inequality above inside (10). □

We now dispose of all the necessary results to prove Theorem B.10.

*Proof.* Let $u \in \mathcal{S}_i$. Let $\delta \in (0, 1)$. By invoking Lemma D.2, then Lemma D.3 then finally Lemma D.1, we find that, with probability $1 - \delta/|\mathcal{S}_i|$

$$\mathcal{R}\left(p_i^{1:T}, c^{1:T}; u\right) \leq \frac{5m}{\gamma_T} + 2\sum_{t=1}^{T} \gamma_t \|\hat{c}^t\|_2^2 + 2mc_{\max}\sum_{t=1}^{T} \mu_t + m^3 c_{\max} \vartheta^{3/2} \sqrt{\sum_{t=1}^{T} \frac{1}{\mu_t^2} \log(|\mathcal{S}_i|/\delta)}$$

$$+ c_{\max} m \sqrt{T \log\left(\frac{|\mathcal{S}_i|}{\delta}\right)} + 32nm^4$$

By invoking Lemma E.1,

$$\mathcal{R}\left(p_i^{1:T}, c^{1:T}; u\right) \leq \frac{5m}{\gamma_T} + 2\sum_{t=1}^{T} \frac{\gamma_t m^5 c_{\max}^2 \vartheta^2}{\mu_t^2} + 2mc_{\max}\sum_{t=1}^{T} \mu_t + m^3 c_{\max} \vartheta^{3/2} \sqrt{\sum_{t=1}^{T} \frac{1}{\mu_t^2} \log(|\mathcal{S}_i|/\delta)}$$

$$+ c_{\max} m \sqrt{T \log\left(\frac{|\mathcal{S}_i|}{\delta}\right)} + 32nm^4$$

Now plugging in the choice of step-sizes $\gamma_t = \sqrt{\frac{c_{\max}\mu_t}{\vartheta n^3 m^3 t}}$ and $\mu_t = \frac{m^{4/5} n^{1/5} \vartheta^{1/5}}{t^{1/5} c_{\max}^{1/5}}$, we have that

$$\mathcal{R}\left(p_i^{1:T}, c^{1:T}; u\right) \leq \tilde{\mathcal{O}}\left(m^{2.3} c^{2.8} \sqrt{\log \frac{|\mathcal{S}_i|}{\delta}} T^{4/5}\right)$$

Finally, using a union bound, the regret above holds uniformly for any $u \in \mathcal{S}_i$ with probability $1 - \delta$. In particular it holds for the fixed strategy in hindsight. Consequently,

$$\mathcal{R}\left(p_i^{1:T}, c^{1:T}\right) \leq \tilde{\mathcal{O}}\left(m^{2.8} c^{2.8} T^{4/5} \sqrt{\log \frac{1}{\delta}}\right)$$

where we have used the fact that $\log|\mathcal{S}_i| \leq m$. □

*Remark* F.4. Notice that the choice of $\gamma_t$ and $\mu_t$ are done to optimize the rate of convergence to NE. To optimize the regret bound, we can choose $\gamma_t = \frac{\mu_t}{m^2 c_{\max} \vartheta t}$ and $\mu_t = \frac{1}{2t^{1/4}}$ to obtain $\mathcal{R}\left(p_i^{1:T}, c^{1:T}\right) \leq m^3 T^{3/4}$.

# G. Nash convergence analysis

## G.1. Properties of the potential function $\Phi$

In this section we show that the potential function is bounded, Lipschitz and smooth. All three properties will be used in later proofs. Recall that the potential function is given by

$$\Phi(x) = \sum_{e \in E} \sum_{\mathcal{S} \subseteq [n]} \prod_{j \in \mathcal{S}} x_{je} \prod_{j \notin \mathcal{S}} (1 - x_{je}) \sum_{\ell=0}^{|\mathcal{S}|} c_e(\ell)$$

**Lemma G.1** (Bounded potential function). *The potential function $\Phi$ is bounded and for all $x \in \mathcal{X}$,*

$$|\Phi(x)| \leq nmc_{\max}$$

*Proof.* This can easiliy be seen by rewriting the potential function as follows

$$\Phi(x) = \sum_{e \in E} \sum_{\mathcal{S} \subseteq [n]} \prod_{j \in \mathcal{S}} x_{je} \prod_{j \notin \mathcal{S}} (1 - x_{je}) \sum_{\ell=0}^{|\mathcal{S}|} c_e(\ell)$$

$$= \sum_{e \in E} \sum_{\mathcal{S} \subseteq [n]} \mathbb{P} \left(\text{``set of agents that picked } e\text{''} = \mathcal{S}\right) \sum_{\ell=0}^{|\mathcal{S}|} c_e(\ell)$$

$$\leq nc_{\max} \sum_{e \in E} \sum_{\mathcal{S} \subseteq [n]} \mathbb{P} \left(\text{``set of agents that picked } e\text{''} = \mathcal{S}\right)$$

$$= nc_{\max} \sum_{e \in E} 1$$

$$= nmc_{\max}$$

$\square$

**Lemma G.2** (Lipschitz potential function). *The gradient of $\Phi$ is bounded and*

$$\|\nabla \Phi(x)\|_2 \leq \sqrt{nm} c_{\max}$$

*Proof.* We start my computing the gradient coordinate at $i, e$ for $i \in [n]$ and $e \in [m]$.

$$\frac{\partial \Phi(x)}{\partial x_{ie}} = \sum_{\mathcal{S}_{-i} \subseteq [n-1]} \prod_{j \in \mathcal{S}_{-i}} x_{je} \prod_{j \notin \mathcal{S}_{-i}} (1 - x_{je}) \sum_{\ell=0}^{|\mathcal{S}_{-i}|+1} c_e(\ell) - \sum_{\mathcal{S}_{-i} \subseteq [n-1]} \prod_{j \in \mathcal{S}_{-i}} x_{je} \prod_{j \notin \mathcal{S}_{-i}} (1 - x_{je}) \sum_{\ell=0}^{|\mathcal{S}_{-i}|} c_e(\ell) \quad (11)$$

$$= \sum_{\mathcal{S}_{-i} \subseteq [n-1]} \prod_{j \in \mathcal{S}_{-i}} x_{je} \prod_{j \notin \mathcal{S}_{-i}} (1 - x_{je}) c_e \left(|\mathcal{S}_{-i}| + 1\right). \quad (12)$$

Observe then that

$$0 \leq \frac{\partial \Phi(x)}{\partial x_{ie}} \leq c_{\max}$$

Since the $\ell_\infty$ norm is bounded by $c_{\max}$, we obtain the $\ell_2$ norm bound by multiplying by the dimension. $\square$

**Lemma G.3** (Smooth potential function). *(Lemma 9 of (Panageas et al., 2023)) The gradient of $\Phi$ is Lipschitz continuous and for any $x, y \in \mathcal{X}$*

$$\|\nabla \Phi(x) - \nabla \Phi(y)\| \leq 2n^2 \sqrt{m} c_{\max} \|x - y\|_2$$

With this lemma, proving that $\tilde{\Phi}$ is smooth becomes immediate.

**Proposition G.4.** *The function $\tilde{\Phi}$ is $\frac{1}{\lambda}$-smooth with $\lambda = (2n^2 m^{7/2} c_{\max})^{-1}$.*

*Proof.* The operator norm of the matrix $B$ can easily be bounded as it is a block diagonal matrix. Indeed we have that

$$\|B\|_2 \leq \max_{i=1,\ldots,n} \|B_i\|_2 \leq \max_{i=1,\ldots,n} \|B_i\|_F \leq m^2.$$

Conseqently, the smoothness constant of $\tilde{\Phi}$ is obtained by multiplying the smoothness constant of $\Phi$ by $m^2$. $\quad\square$

A final property we will use is the following which states that if all other players stay fixed, the cost incurred by a single agent $i$ is linear in terms of its strategy.

**Lemma G.5** (Linearized cost). *Let $\pi \in \Delta(\mathcal{S}_1) \times \ldots \Delta(\mathcal{S}_n)$ with marginalization $x \in \mathcal{X}$. Then, for all $i \in [n]$,*

$$C_i(\pi_i, \pi_{-i}) = \left\langle \frac{\partial \Phi(x)}{\partial x_i}, x_i \right\rangle$$

*and $\frac{\partial \Phi(x)}{\partial x_i}$ only depends on $x_{-i}$.*

*Proof.* Let $i \in [n]$. By definition of the cost,

$$C_i(\pi_i, \pi_{-i}) = \mathbb{E}_{(p_i, p_{-i}) \sim (\pi_i, \pi_{-i})} \left[ \sum_{e \in p_i} c_e(\ell_e(p_i, p_{-i})) \right]$$

$$= \mathbb{E}_{p_i \sim \pi_i} \left[ \mathbb{E}_{p_{-i} \sim \pi_{-i}} \left[ \sum_{e \in E} c_e(\ell_e(p_i, p_{-i})) \mathbb{1}\left[ e \in p_i \right] \middle| p_i \right] \right]$$

$$= \sum_{e \in E} \mathbb{E}_{p_{-i} \sim \pi_{-i}} \left[ c_e(\ell_e(p_{-i}) + 1) \right] \mathbb{E}_{p_i \sim \pi_i} \left[ \mathbb{1}\left[ e \in p_i \right] \right]$$

$$= \sum_{e \in E} \mathbb{E}_{p_{-i} \sim \pi_{-i}} \left[ c_e(\ell_e(p_{-i}) + 1) \right] x_{ie}$$

where the third equality follows form the fact that $c_e(\ell_e(p_i, p_{-i})) \mathbb{1}\left[ e \in p_i \right] = c_e(\ell_e(p_{-i}) + 1) \mathbb{1}\left[ e \in p_i \right]$). We then observe that $\mathbb{E}_{p_{-i} \sim \pi_{-i}} \left[ c_e(\ell_e(p_{-i}) + 1) \right]$ is precisely what is computed in equation (12) to find that

$$C_i(\pi_i, \pi_{-i}) = \left\langle \frac{\partial \Phi(x)}{\partial x_i}, x_i \right\rangle$$

$\quad\square$

## G.2. Proof of Theorem B.11

As stated in section D.2, we show convergence to Nash equilibria by showing convergence to a stationary point of the potential function. This strategy is valid because of the following result relating Nash equilibria with stationary points.

**Proposition G.6** (From Stationarity to Nash). *Let $\pi \in \Delta(\mathcal{S}_1) \times \cdots \times \Delta(\mathcal{S}_n)$. Let $x \in \mathcal{X}$ be the marginalization of $\pi$. If $x = B\alpha$, with $\alpha \in \mathcal{D}$ an $(\epsilon, \mu)$-stationary point, then $\pi$ is a $4n^{2.5}m^4 c_{\max}(\epsilon + \mu)$-mixed Nash equilibrium.*

*Proof.* Let $\pi'_i \in \Delta(\mathcal{X}_i)$ with marginalization $x'_i \in \mathcal{X}_i$. Let $x' = [x_1, \ldots, x'_i, \ldots, x_n]$ differ from $x$ only at $x'_i$. By definition of the potential function, we know that

$$C_i(\pi_i, \pi_{-i}) - C_i(\pi'_i, \pi_{-i}) = \Phi(x_i, x_{-i}) - \Phi(x'_i, x_{-i})$$

By further invoking Lemma G.5, and using the fact that $\frac{\partial \Phi(x)}{\partial x_i}$ only depends on $x_{-i}$, we have that

$$C_i(\pi_i, \pi_{-i}) - C_i(\pi'_i, \pi_{-i}) = \left\langle \frac{\partial \Phi(x)}{\partial x_i}, x_i - x'_i \right\rangle = \langle \nabla \Phi(x), x - x' \rangle$$

where the last equality comes from the fact that $x - x'$ is zero except on the $x_i$ block of coordinates. Since $x - x' = B(\alpha - \alpha')$ for some $\alpha' \in \mathcal{D}$, we have that

$$C_i(\pi_i, \pi_{-i}) - C_i(\pi'_i, \pi_{-i}) = \left\langle \nabla \tilde{\Phi}(x), \alpha - \alpha' \right\rangle$$

We now exploit the fact that $\alpha$ is stationary. Let $\alpha^+ = \Pi_{\mathcal{D}^\mu}\left[\alpha - \frac{\lambda}{2}\tilde{\Phi}(\alpha)\right]$. By definition of the projection, for any $u \in \mathcal{D}^\mu$, it holds that

$$\left\langle \alpha - \frac{\lambda}{2}\nabla\tilde{\Phi}(\alpha) - \alpha^+, u - \alpha^+ \right\rangle \le 0$$

By rearranging, we find that

$$\left\langle \nabla\tilde{\Phi}(\alpha), \alpha^+ - u \right\rangle \le \frac{2}{\lambda}\left\langle \alpha - \alpha^+, \alpha^+ - u \right\rangle$$

With this inequality in hand, we obtain that

$$\begin{aligned}
\left\langle \nabla\tilde{\Phi}(\alpha), \alpha - u \right\rangle &= \left\langle \nabla\tilde{\Phi}(\alpha), \alpha^+ - u \right\rangle + \left\langle \nabla\tilde{\Phi}(x), \alpha - \alpha^+ \right\rangle \\
&\le \frac{2}{\lambda}\left\langle \alpha - \alpha^+, \alpha^+ - u \right\rangle + \left\langle \nabla\tilde{\Phi}(\alpha), \alpha - \alpha^+ \right\rangle \\
&\le \left( \frac{2\sqrt{nm}}{\lambda} + \|\nabla\tilde{\Phi}(\alpha)\|_2 \right) \|\alpha^+ - \alpha\|_2 \\
&\le \left( 4n^{2.5}m^4 c_{\max} \right) G^\mu(\alpha).
\end{aligned}$$

To conclude we simply take $u = (1-\mu)\alpha' + \mu\frac{1}{s}\mathbb{1}$ which is necessarily in $\mathcal{D}^\mu$ to find that

$$\begin{aligned}
\left\langle \nabla\tilde{\Phi}(x), x - x' \right\rangle &= \left\langle \nabla\tilde{\Phi}(x), x - u \right\rangle + \left\langle \nabla\tilde{\Phi}(x), u - x' \right\rangle \\
&\le \left( 4n^{2.5}m^4 c_{\max} \right) G^\mu(x) + nmc_{\max}\mu \\
&\le 4n^{2.5}m^4 c_{\max} \left( G^\mu(x) + \mu \right)
\end{aligned}$$

$\square$

Thanks to the proposition above we can focus our attention on proving convergence to stationary points.

**Lemma G.7** (Estimator property). *Let $t \in [T]$ and $\mathcal{F}_t$ be the sigma-field generated by $\alpha_1, \ldots, \alpha_t$ and denote the conditional expectation as $\mathbb{E}_t[\cdot] \triangleq \mathbb{E}[\cdot|\mathcal{F}_t]$. It holds that*

1. $\mathbb{E}_t[\nabla_t] = \nabla\tilde{\Phi}(\alpha^t)$,

2. $\mathbb{E}_t[\|\nabla_t\|_2^2] \le \frac{nm^4 c_{\max}^2}{\mu_t}$

*Proof.* Let $i \in [n]$ and $e \in E$. First, observe that from lemma G.5, we have that the linearized cost $c^t$ for agent $i$ satisfies

$$\mathbb{E}_t\left[c_e^t\right] = \frac{\partial\Phi}{\partial x_{ie}}(x^t)$$

Now using the tower property, we have that

$$\begin{aligned}
\mathbb{E}_t[[\nabla_t]_i] = \mathbb{E}_t\left[B_i^\top \hat{c}_i^t\right] &= B_i^\top \mathbb{E}_t\left[\mathbb{E}\left[M_{i,t}^+ p_i^t \left(\sum_{e \in p_i^t} c_e^t\right) |p_i^t\right]\right] \\
&= B_i^\top \sum_{p_k \in \mathrm{supp}(\pi_i^t)} \mathbb{P}\left(p_i^t = p_k\right) M_{i,t}^+ p_k \sum_{e \in p^k} \mathbb{E}_t\left[c_e^t | p_i^t = p_k\right] \\
&= B_i^\top \sum_{p_k \in \mathrm{supp}(\pi_i^t)} \mathbb{P}\left(p_i^t = p_k\right) M_{i,t}^+ p_k \sum_{e \in p^k} \frac{\partial\Phi}{\partial x_{ie}}(x^t) \\
&= B_i^\top \sum_{p_k \in \mathrm{supp}(\pi_i^t)} \mathbb{P}\left(p_i^t = p_k\right) M_{i,t}^+ p_k p_k^T \frac{\partial\Phi}{\partial x_i}(x^t) \\
&= B_i^\top M_{i,t}^+ M_{i,t} \frac{\partial\Phi}{\partial x_i}(x^t) \\
&= B_i^\top \frac{\partial\Phi}{\partial x_i}(x^t)
\end{aligned}$$

where the last equality follows from (8). We thus conclude that

$$\mathbb{E}_t\left[\nabla_t\right] = \nabla\tilde{\Phi}(\alpha^t).$$

For the second point, we know from equation (9) in the proof of Lemma E.1 that

$$B_i^\top \hat{c}^t = \left\langle c^t, p_i^t \right\rangle N_{i,t}^+ \alpha_{i,t}^p \tag{13}$$

We can then control the expectation of square norm of this estimator as follows

$$
\begin{aligned}
\mathbb{E}_t\left[\|B_i^\top \hat{c}^t\|_2^2\right] &\leq m^2 c_{\max}^2 \mathbb{E}_t\left[\left\|N_{i,t}^+ \alpha_{i,t}^p\right\|_2^2\right]\\
&= m^2 c_{\max}^2 \mathbb{E}_t\left[\operatorname{tr}\left(N_{i,t}^+ \alpha_{i,t}^p \alpha_{i,t}^{p\top} N_{i,t}^{+\top}\right)\right]\\
&= m^2 c_{\max}^2 \operatorname{tr}\left(N_{i,t}^+ \mathbb{E}_t\left[\alpha_{i,t}^p \alpha_{i,t}^{p\top}\right] N_{i,t}^{+\top}\right)\\
&\leq m^2 c_{\max}^2 \operatorname{tr}\left(N_{i,t}^+\right)\\
&\leq m^4 c_{\max}^2 \frac{1}{\mu_t}
\end{aligned}
$$

where the last inequality follows from (7) where we have used that $s \leq m$. Now, since $\nabla_t$ is a concatenation of the estimators $B_i^\top \hat{c}^t$, we find that

$$\mathbb{E}_t\left[\|\nabla_t\|_2^2\right] \leq \frac{nm^4 c_{\max}^2}{\mu_t}.$$

$\square$

**Lemma G.8** (Gap control). *Let $G^t(\alpha) := \|\Pi_{\mathcal{D}^{\mu_t}}\left[\alpha - \frac{\lambda}{2}\nabla\tilde{\Phi}(\alpha)\right] - x\|_2$ denote the $\mu_t$-stationarity gap. We have that for any $\alpha \in \mathcal{D}^{\mu_t}$,*

$$G^t(\alpha) \leq \lambda\|\nabla M_{\lambda\tilde{\Phi}}^t(\alpha)\|_2$$

*Proof.* The proof relies on introducing a fixed point $y$ such that

$$y = \Pi_{\mathcal{D}^\mu}\left[x - \frac{\lambda}{2}\nabla\tilde{\Phi}(y)\right].$$

Luckily the point $y = x - \frac{\lambda}{2}\nabla M_{\lambda\tilde{\Phi}}^\mu(x)$ is such a fixed point(see point 2 in I.2). Now we can write

$$
\begin{aligned}
G^\mu(x) &= \|\Pi_{\mathcal{D}^\mu}\left[x - \frac{\lambda}{2}\nabla\tilde{\Phi}(x)\right] - x\|_2\\
&\leq \|\Pi_{\mathcal{D}^\mu}\left[x - \frac{\lambda}{2}\nabla\tilde{\Phi}(x)\right] - \Pi_{\mathcal{D}^\mu}\left[x - \frac{\lambda}{2}\nabla\tilde{\Phi}(y)\right]\|_2 + \|y - x\|_2\\
&\leq \frac{\lambda}{2}\|\nabla\tilde{\Phi}(x) - \nabla\tilde{\Phi}(y)\| + \|y - x\|_2\\
&\leq \frac{3}{2}\|y - x\|_2 = \frac{3\lambda}{4}\|\nabla M_{\lambda\tilde{\Phi}}^\mu(x)\|_2 \leq \lambda\|\nabla M_{\lambda\tilde{\Phi}}^\mu(x)\|_2
\end{aligned}
$$

$\square$

**Theorem D.11** (Stochastic gradient descent). *Consider the sequence $\alpha^1, \ldots, \alpha^T$ produced by Equation 6. Then,*

$$\frac{1}{T}\sum_{t=1}^T \mathbb{E}\left[\|\nabla M_{\lambda\tilde{\Phi}}^t(\alpha^t)\|_2\right] \leq 2n^{1.5}\sqrt{\frac{2m^{1.5}c_{\max}}{\gamma_T T} + \frac{n^3 m^{7.5}}{\gamma_T T}\sum_{t=1}^T \frac{\gamma_t^2}{\mu_t}}$$

*Proof.* Let us first recall some of the notation we use. The time dependent Moreau envelope is given by

$$M^t_{\lambda\tilde{\Phi}}(x) \triangleq \min_{y \in \mathcal{D}^{\mu_t}} \left\{ \tilde{\Phi}(y) + \frac{1}{\lambda} \|x - y\|_2^2 \right\},$$

Notice here that the envelope is taken with respect to a time varying polytope. The iterates $\alpha^{1:T}$ are updated by the following update rule

$$\alpha^{t+1} = \Pi_{\mathcal{D}^{\mu_{t+1}}} \left[ \alpha^t - \gamma_t \cdot \nabla_t \right] \tag{14}$$

With this in mind, we proceed with the proof. Since $M^t_{\lambda\tilde{\Phi}}$ is $\frac{2}{\lambda}$-smooth (by point 4 of Lemma I.2), we have that

$$M^t_{\lambda\tilde{\Phi}}(\alpha^{t+1}) \le M^t_{\lambda\tilde{\Phi}}(\alpha^t) + \left\langle \nabla M^t_{\lambda\tilde{\Phi}}(\alpha^t), \alpha^{t+1} - \alpha^t \right\rangle + \frac{1}{\lambda} \|\alpha^{t+1} - \alpha^t\|_2^2$$

Now since $\nabla M^t_{\lambda\tilde{\Phi}}(\alpha^t) = \frac{2}{\lambda} \left( \alpha^t - \text{prox}^t_{\frac{\lambda}{2}\tilde{\Phi}}(\alpha^t) \right)$ (by point 3 of Lemma I.2), where we can invoke the contractive properties of the projection in (14) to find that

$$M^t_{\lambda\Phi}(\alpha^{t+1}) \le M^t_{\lambda\Phi}(\alpha^t) - \gamma_t \left\langle \nabla M^t_{\lambda\tilde{\Phi}}(\alpha^t), \nabla_t \right\rangle + \frac{\gamma_t^2}{\lambda} \|\nabla_t\|_2^2$$

Taking the expectation, we have

$$\mathbb{E}\left[ M^t_{\lambda\Phi}(\alpha^{t+1}) \right] \le \mathbb{E}\left[ M^t_{\lambda\Phi}(\alpha^t) \right] - \gamma_t \mathbb{E}\left[ \left\langle \nabla M^t_{\lambda\tilde{\Phi}}(\alpha^t), \mathbb{E}_t\left[\nabla_t\right] \right\rangle \right] + \frac{\gamma_t^2}{\lambda} \mathbb{E}\left[ \|\nabla_t\|_2^2 \right]$$

Using Lemma D.9, we can replace the terms involving $\nabla_t$ on the right hand side to find that

$$\mathbb{E}\left[ M^t_{\lambda\Phi}(\alpha^{t+1}) \right] \le \mathbb{E}\left[ M^t_{\lambda\Phi}(\alpha^t) \right] - \gamma_t \mathbb{E}\left[ \left\langle \nabla M^t_{\lambda\tilde{\Phi}}(\alpha^t), \nabla\tilde{\Phi}(\alpha^t) \right\rangle \right] + \frac{nm^4 c_{\max}^2}{\lambda} \frac{\gamma_t^2}{\mu_t}$$

Invoking Lemma G.9, we obtain

$$\mathbb{E}\left[ M^t_{\lambda\Phi}(\alpha^{t+1}) \right] \le \mathbb{E}\left[ M^t_{\lambda\Phi}(\alpha^t) \right] - \frac{\gamma_t}{4} \|\nabla M^t_{\lambda\tilde{\Phi}}(\alpha^t)\|_2^2 + \frac{nm^4 c_{\max}^2}{\lambda} \frac{\gamma_t^2}{\mu_t}$$

By rearranging the terms, we can write that

$$\frac{\gamma_t}{4} \mathbb{E}\left[ \|\nabla M^t_{\lambda\tilde{\Phi}}(\alpha^t)\|_2^2 \right] \le \mathbb{E}\left[ M^t_{\lambda\Phi}(\alpha^t) \right] - \mathbb{E}\left[ M^t_{\lambda\Phi}(\alpha^{t+1}) \right] + \frac{nm^4 c_{\max}^2}{\lambda} \frac{\gamma_t^2}{\mu_t}$$

At this point we notice that $M^{t+1}_{\lambda\tilde{\Phi}}(\alpha^{t+1}) \le M^t_{\lambda\tilde{\Phi}}(\alpha^{t+1})$ since $\mathcal{D}^{\mu_t} \subset \mathcal{D}^{\mu_{t+1}}$, which gives us

$$\frac{\gamma_t}{4} \mathbb{E}\left[ \|\nabla M^t_{\lambda\tilde{\Phi}}(\alpha^t)\|_2^2 \right] \le \mathbb{E}\left[ M^t_{\lambda\Phi}(\alpha^t) \right] - \mathbb{E}\left[ M^{t+1}_{\lambda\Phi}(\alpha^{t+1}) \right] + \frac{nm^4 c_{\max}^2}{\lambda} \frac{\gamma_t^2}{\mu_t}$$

Now summing from $t = 1, \ldots, T$ and telescoping, we find that

$$\frac{1}{T} \sum_{t=1}^{T} \mathbb{E}\left[ \|\nabla M^t_{\lambda\tilde{\Phi}}(\alpha^t)\|_2^2 \right] \le \frac{8 M^{\max}_{\lambda\tilde{\Phi}}}{\gamma_T T} + 4 \frac{nm^4 c_{\max}^2}{\lambda \gamma_T T} \sum_{t=1}^{T} \frac{\gamma_t^2}{\mu_t}$$

where we have used the fact that $\gamma_T \le \gamma_t$ and defined $M^{\max}_{\lambda\tilde{\Phi}} := \max_{t \in [T]} \max_{x \in \mathcal{D}^{\mu_t}} M^t_{\lambda\tilde{\Phi}}(x)$. By taking the square root and applying Jensen's inequality, we have that

$$\frac{1}{T} \sum_{t=1}^{T} \mathbb{E}\left[ \|\nabla M^t_{\lambda\tilde{\Phi}}(\alpha^t)\|_2 \right] \le \sqrt{\frac{8 M^{\max}_{\lambda\tilde{\Phi}}}{\gamma_T T} + 4 \frac{nm^4 c_{\max}^2}{\lambda \gamma_T T} \sum_{t=1}^{T} \frac{\gamma_t^2}{\mu_t}}$$

Finally by plugging in the values of $M^{\max}_{\lambda\tilde{\Phi}} \le n^3 m^{3/2} c_{\max}$ and $\frac{1}{\lambda} = 2n^2 m^{7/2} c_{\max}$, we find that

$$\frac{1}{T} \sum_{t=1}^{T} \mathbb{E}\left[ \|\nabla M^t_{\lambda\tilde{\Phi}}(\alpha^t)\|_2 \right] \le 2n^{1.5} \sqrt{\frac{2m^{1.5} c_{\max}}{\gamma_T T} + \frac{\vartheta n^3 m^{7.5}}{\gamma_T T} \sum_{t=1}^{T} \frac{\gamma_t^2}{\mu_t}}$$

$\square$

**Lemma G.9.** *For any $t \in [T]$, we have that*

$$\left\langle \nabla M_{\lambda\tilde\Phi}^t(\alpha^t), \nabla\tilde\Phi(\alpha^t) \right\rangle \geq \frac{1}{4}\|\nabla M_{\lambda\tilde\Phi}^t(\alpha^t)\|_2^2$$

*Proof.* This lemma is obtained by exploiting the smoothness of $\Phi$. We begin by defining the gradient step $y^t := \alpha^t - \frac{\lambda}{2}\nabla M_{\lambda\tilde\Phi}^t(\alpha^t)$, which allows us to write

$$\left\langle \nabla M_{\lambda\tilde\Phi}^t(\alpha^t), \nabla\tilde\Phi(\alpha^t) \right\rangle = -\frac{2}{\lambda}\left\langle y^t - \alpha^t, \nabla\tilde\Phi(\alpha^t) \right\rangle. \tag{15}$$

Now since $\Phi$ is $\frac{1}{\lambda}$-smooth, we have that

$$
\begin{aligned}
-\left\langle y^t - \alpha^t, \nabla\tilde\Phi(\alpha^t) \right\rangle &\geq \tilde\Phi(\alpha^t) - \tilde\Phi(y^t) - \frac{1}{2\lambda}\|y^t - \alpha^t\|_2^2 \\
&= \left(\tilde\Phi(\alpha^t) + \frac{1}{\lambda}\|\alpha^t - \alpha^t\|_2^2\right) - \left(\tilde\Phi(y^t) + \frac{1}{\lambda}\|y^t - \alpha^t\|_2^2\right) + \frac{1}{2\lambda}\|y^t - \alpha^t\|_2^2 \\
&\geq \frac{1}{2\lambda}\|y^t - \alpha^t\|_2^2 \quad \text{(because } y^t = \underset{y \in \mathcal{D}_i^{\mu_{t+1}}}{\arg\min}\ \tilde\Phi(y) + \frac{1}{\lambda}\|\alpha^t - y\|_2^2\text{)} \\
&= \frac{\lambda}{8}\|\nabla M_{\lambda\tilde\Phi}^t(\alpha^t)\|_2^2.
\end{aligned}
$$

Plugging this result into (15) gives

$$\left\langle \nabla M_{\lambda\tilde\Phi}^t(\alpha^t), \nabla\tilde\Phi(\alpha^t) \right\rangle \geq \frac{1}{4}\|\nabla M_{\lambda\tilde\Phi}^t(\alpha^t)\|_2^2.$$

$\square$

We can now proceed to prove Theorem B.11.

*Proof.* Let $u$ be sampled uniformly from $[T]$. The joint strategy profile $\pi^u$ has marginalization $\alpha^u \in \mathcal{D}^{\mu_u}$, and therefore, by lemma D.8 we have that

$$\frac{1}{T}\mathbb{E}\left[\sum_{t=1}^T \max_{i\in[n]}\left[c_i(\pi_i^t, \pi_{-i}^t) - \min_{\pi_i \in \Delta(\mathcal{P}_i)} c_i(\pi_i, \pi_{-i}^t)\right]\right] \leq 4n^{2.5}m^4 c_{\max}\mathbb{E}\left[G^u(x^u) + \mu_u\right]$$

Expanding the right hand side, we have that

$$\mathbb{E}\left[G^u(x^u) + \mu_u\right] \leq \frac{1}{T}\sum_{t=1}^T \mathbb{E}\left[G^t(x^t)\right] + \frac{1}{T}\sum_{t=1}^T \mu_t$$

By Lemma D.10, we get that

$$\mathbb{E}\left[G^u(x^u) + \mu_u\right] \leq \frac{\lambda}{T}\sum_{t=1}^T \mathbb{E}\left[\|\nabla M^t(x^t)\|_2\right] + \frac{1}{T}\sum_{t=1}^T \mu_t$$

It then follows by Theorem D.11 that

$$
\begin{aligned}
\mathbb{E}\left[G^u(x^u) + \mu_u\right] &\leq 2\lambda n^{1.5}\sqrt{\frac{2m^{1.5}c_{\max}}{\gamma_T T} + \frac{n^3 m^{7.5}}{\gamma_T T}\sum_{t=1}^T \frac{\gamma_t^2}{\mu_t} + \frac{1}{T}\sum_{t=1}^T \mu_t} \\
&= \frac{1}{\sqrt{n}m^4 c_{\max}}\sqrt{\frac{2m^{1.5}c_{\max}}{\gamma_T T} + \frac{n^3 m^{7.5}}{\gamma_T T}\sum_{t=1}^T \frac{\gamma_t^2}{\mu_t} + \frac{1}{T}\sum_{t=1}^T \mu_t}
\end{aligned}
$$

Now, plugging in $\gamma_t = \sqrt{\frac{c_{\max}\mu_t}{n^3 m^6 t}}$

$$\mathbb{E}\left[G^u(x^u) + \mu_u\right] \leq \frac{1}{\sqrt{n}m^4 c_{\max}}\sqrt{\frac{c_{\max}^{1.5}m^{4.5}n^{1.5}\log T}{\sqrt{T}\mu_T}} + \frac{1}{T}\sum_{t=1}^{T}\mu_t$$

$$\leq \frac{n^{1/4}}{m^{1.75}c_{\max}^{1/4}}\sqrt{\frac{3\log T}{\sqrt{T}\mu_T}} + \frac{1}{T}\sum_{t=1}^{T}\mu_t$$

Finally, setting the exploration parameter $\mu_t = \frac{n^{1/5}}{m^{7/5}t^{1/5}c_{\max}^{1/5}}$ and using the fact that $\sum_{t=1}^{T}t^{-1/5} \leq \frac{5T^{4/5}}{4}$, we obtain

$$\frac{1}{T}\mathbb{E}\left[\sum_{t=1}^{T}\max_{i\in[n]}\left[c_i(\pi_i^t, \pi_{-i}^t) - \min_{\pi_i\in\Delta(\mathcal{P}_i)}c_i(\pi_i, \pi_{-i}^t)\right]\right] \leq \frac{4m^{2.6}n^{2.7}c_{\max}^{4/5}}{T^{1/5}}.$$

Therefore choosing $T \geq \frac{4^5 m^{13} n^{13.5} c_{\max}^4}{\epsilon}$ ensures

$$\frac{1}{T}\mathbb{E}\left[\sum_{t=1}^{T}\max_{i\in[n]}\left[c_i(\pi_i^t, \pi_{-i}^t) - \min_{\pi_i\in\Delta(\mathcal{P}_i)}c_i(\pi_i, \pi_{-i}^t)\right]\right] \leq \epsilon$$

$\square$

We now have all the ingredients we need to prove Corollary 2.7.

*Proof.* Let $u$ be sampled uniformly from $[T]$. The joint strategy profile $\pi^u$ has marginalization $\alpha^u \in \mathcal{D}^{\mu_u}$, and therefore, by lemma D.8, it is a

$$4n^{2.5}m^4 c_{\max}\left(G^u(x^u) + \mu_u\right) - \text{mixed Nash equilibrium}$$

Now let $\delta \in (0,1)$. By Markov's inequality and Theorem B.11,

$$\max_{i\in[n]}\left[c_i(\pi_i^u, \pi_{-i}^u) - \min_{\pi_i\in\Delta(\mathcal{P}_i)}c_i(\pi_i, \pi_{-i}^u)\right] \leq \epsilon/\delta$$

with probability $1 - \delta$ if $T \geq \frac{4^5 m^{13} n^{13.5} c_{\max}^4 \theta}{\epsilon}$. Finally, putting everything together we find that $\pi^u$ is a

$$\tilde{\mathcal{O}}\left(\frac{n^{2.7}m^{13/5}c_{\max}^{4/5}}{\delta}T^{-1/5}\right)$$

with probability $1 - \delta$. Finally, to make the quantity $\frac{n^{2.7}m^{13/5}c_{\max}^{4/5}}{\delta}T^{-1/5}$ equal to $\epsilon/\delta$ we choose $T \geq \Theta\left(m^{13}n^{13.5}/\epsilon^5\right)$.

For the first statement of the corollary, we the set of time steps $\mathcal{B} := \{t \in \{1, t\} : E_t > \epsilon/\delta^2\}$ where $E_t := \max_{i\in[n]}\left[c_i(\pi_i^t, \pi_{-i}^t) - \min_{\pi_i\in\Delta(\mathcal{P}_i)}c_i(\pi_i, \pi_{-i}^t)\right]$ which is a random variable. With probability $1 - \delta$, $\sum_{t=1}^{T}E_t \leq \frac{\epsilon T}{\delta}$ we directly get that we probability $1 - \delta$, $|\mathcal{B}| \leq \delta T$. As a result, with probability $\geq 1 - \delta$, $(1 - \delta)$ fraction of the profiles $\pi^1, \ldots, \pi^T$ are $\epsilon/\delta^2$-Mixed NE. $\square$

## H. Spanner construction omitted proofs

*Proof of C.7.* We proceed by induction on the topological ordering. For $v_{n-1}$, we pick a redundant outgoing edge. By definition of a topological ordering, the chosen edge will necessarily lead to $v_n = t_i$.

Now let $k \in [2, n-2]$ and assume that the lemma holds for all for $l > k$. We consider the node $v_k$ and pick an outgoing redundant edge. It will lead to a node $v_l$ with $l > k$. By induction hypothesis, there exists a path connecting $v_l$ to $t_i$ that only consists of redundant edges. Concatenating the picked outgoing edge with this path yields the result for $v_k$ so the lemma holds for $k$. $\square$

*Proof of C.8.* Suppose $i = \texttt{Prefix}(k) = \texttt{Prefix}(l)$. Then by construction $e_i < e_k < e_l$. On the other hand, since the prefixes are set in reverse topological order and $e_k$ and $e_l$ are connected, we must have $\texttt{Prefix}(l) \geq k$. A contradiction. $\square$

## I. Technical Lemmas

**Lemma I.1** (Projection lemma). *Let $\mathcal{D}_i^\mu$ be a bounded away polytope. For any $z \in \mathbb{R}^s$, the projection on $\mathcal{D}_i^\mu$ can be expressed as*

$$\Pi_{\mathcal{D}_i^\mu}[z] = (1 - \mu)\Pi_{\mathcal{D}_i}\left[\frac{1}{1-\mu}\left(z - \frac{\mu}{s}\mathbb{1}\right)\right] + \frac{\mu}{s}\mathbb{1}$$

*Proof.* We first express the indicator function of $\mathcal{D}_i^\mu$ in terms of the indicator of $\mathcal{D}_i$. We have that for any $z \in \mathbb{R}^s$, by definition of the bounded away polytope,

$$\iota_{\mathcal{D}_i^\mu}(z) = \iota_{\mathcal{D}_i}\left(\frac{1}{1-\mu}\left(z - \frac{\mu}{s}\mathbb{1}\right)\right), \tag{16}$$

The indicator function of $\mathcal{X}_i^\mu$ is therefore obtained through an affine precomposition of the $\mathcal{X}_i$ indicator. We can determine the prox of an affine precomposition by using properties (i) and (ii) in Table 10.1 of (Combettes & Pesquet, 2011), which yields the simple formula given in equation (2.2) of (Parikh et al., 2014). We thus find that

$$\Pi_{\mathcal{D}_i^\mu}[z] = (1 - \mu)\Pi_{\mathcal{D}_i}\left[\frac{1}{1-\mu}\left(z - \frac{\mu}{s}\mathbb{1}\right)\right] + \frac{\mu}{s}\mathbb{1}$$

$\square$

**Lemma I.2** (Moreau enveloppe and proximity operators). *Let $f : \mathcal{X} \mapsto \mathbb{R}$ be a $1/\lambda$-smooth function. Its Moreau-Yosida regularization defined as*

$$e_\eta f(x) = \inf_{y \in \mathcal{X}} f(y) + \frac{1}{2\eta}\|y - x\|_2^2$$

*verifies the following properties for $\eta < \lambda$,*

*1. The proximity operator given by the equation below is single valued*

$$\mathrm{prox}_{\eta f}(x) = \underset{y \in \mathcal{X}}{\arg\min}\, f(y) + \frac{1}{2\eta}\|y - x\|_2^2. \tag{17}$$

*2. By optimality conditions of (17),*

$$\mathrm{prox}_{\eta f}(x) = \Pi_{\mathcal{X}}\left[x - \eta \nabla f(\mathrm{prox}_{\eta f}(x))\right]$$

*3. $e_\eta f$ is continuously differentiable and*

$$\nabla e_\eta f(x) = \frac{1}{\eta}\left(x - \mathrm{prox}_{\eta f}(x)\right)$$

*4. If $\eta = \lambda/2$, then $\nabla e_\eta f$ is $\frac{1}{\eta}$ smooth.*

*Proof.* All these properties follow from (Hoheisel et al.) Corollary 3.4 because $\frac{1}{\lambda}$ smooth functions are $\frac{1}{\lambda}$ weakly convex functions. In our paper, we work with the function $M_{\lambda\tilde{\Phi}}$, notice that it corresponds to the Moreau-Yosida regularization

$$M_{\lambda\Phi} = e_{\frac{\lambda}{2}}\tilde{\Phi}$$

All the properties therefore follow with $\eta = \frac{\lambda}{2}$. $\square$

**Lemma I.3** (Telescoping Lemma). *Let $(\gamma_t)_t$ be a non-increasing sequence. Let $(u_t)_t \in \mathbb{R}_+^{\mathbb{N}}$ be a non-negative sequence uniformly bounded by $u_{\max} > 0$, it holds that*

$$\sum_{t=1}^{T} \frac{1}{\gamma_t}(u_t - u_{t+1}) \leq \frac{u_{\max}}{\gamma_T}$$

*Proof.*

$$\sum_{t=1}^{T} \frac{1}{\gamma_t}(u_t - u_{t+1}) = \sum_{t=1}^{T} \frac{u_t}{\gamma_{t-1}} - \frac{u_{t+1}}{\gamma_t} + \sum_{t=1}^{T} \left( \frac{1}{\gamma_t} - \frac{1}{\gamma_{t-1}} \right) u_t$$

$$\leq \sum_{t=1}^{T} \frac{u_t}{\gamma_{t-1}} - \frac{u_{t+1}}{\gamma_t} + u_{\max} \sum_{t=1}^{T} \frac{1}{\gamma_t} - \frac{1}{\gamma_{t-1}}$$

$$= \frac{u_1}{\gamma_0} - \frac{u_{T+1}}{\gamma_T} + \frac{u_{\max}}{\gamma_T} - \frac{u_{\max}}{\gamma_0}$$

$$\leq \frac{u_{\max}}{\gamma_T}$$

$\square$

