# OpenReview forum: "Polynomial Convergence of Bandit No-Regret Dynamics in Congestion Games"
_ICML.cc/2024/Workshop/Agentic_Markets — Agentic Markets @ ICML'24 Poster_

### Official Review · Reviewer_qSqj · 2024-06-11
**First no-regret and poly(n, m, 1/eps)-round convergent algorithm in congestion games with bandit feedback**

**Rating:** 8
**Confidence:** 3

**Review:**

**Summary**:

This paper studies algorithms for congestion games under bandit feedback (each agent only observes its own cost under the current round’s joint strategy profile).

The authors introduce an algorithm BGD-CE that is no-regret (per-agent regret scaling like O(poly(m) T^0.8)) and converges to an eps-approx NE in poly(n, m, 1/eps) steps when adopted by all agents (where n and m are the number of agents and resources, respectively).

This work answers an open question from Cui et al. (2022), which gave an algorithm converging in poly(n,m, 1/eps) iterations to an eps-approx NE in the bandit feedback model, but without any no-regret guarantee. This also extends the results of Panageas et al. (2023), who gave a no-regret and poly-convergent algorithm in the semi-bandit feedback model, to the more restrictive bandit feedback model.

The algorithm is based on combining online gradient descent with a new exploration component (similar in spirit / an extension of that of Panageas et al.)

**pros**:

+ the paper is well written, with a clear discussion and comparison to the existing regret/NE convergence guarantees of prior works.
+ the new exploration scheme based on barycentric spanners seems novel.

**cons/questions**:
- I wonder if the authors could briefly mention the possible tightness of the regret and NE convergence rates with respect to n and m in this bandit feedback setting?

- The NE convergence notion is in a “best-iterate” sense (similar notion considered in Cui et al. and Panageas et al.). Perhaps the authors could provide a bit more discussion on the connection/differences with last-iterate convergence (and the challenges in establishing such last iterate convergence?).

**small comments**:

- line 144: algorithm called “OGD-CE”, but referred to as “BGD-CE” elsewhere.

---

### Official Review · Reviewer_WPvm · 2024-06-13
**Polynomial Convergence of Bandit No-Regret Dynamics in Congestion Games**

**Rating:** 8
**Confidence:** 4

**Review:**

# Summary of Paper and Contributions
This paper studies an online learning setting in which $n$ agents are participating in a congestion game with $m$ resources. Whilst it is known that best response dynamics converge to a Nash Equilibrium in this setting, the authors argue that BR dynamics have several drawbacks.
Firstly, the convergence guarantees of BR cease to hold when agents update simultaneously. Secondly, BR requires agents to have full information about the state of the congestion game on each time step. Lastly, best response dynamics do not provide regret guarantees for individual agents.
To address this issue, the authors study a specialised setting in which agents only receive bandit feedback, and construct an algorithm which converges to an $\epsilon$-approximate Nash Equilibrium in polynomial time (in both $n$, $m$ and $1/\epsilon$). Moreover the algorithm proposed is no-regret for each individual agent. As a result, the authors address an open question by Cui et al, who asked whether a no-regret bandit algorithm which approximates an $\epsilon$-Nash equilibrium in polynomial time exists.

# Strengths
The paper does a good job of motivating the problem and contextualising their work with respect to existing literature. Moreover, the authors improve upon the technical results of existing work and develop novel, if somewhat incremental, proof techniques. The construction of Barycentric spanners for DAGs may even be of independent interest.

# Weaknesses
The paper spends 75% of the main matter introducing the problem, and as a result little time is dedicated to discussing and conveying the main technical results of the work. In particular, spanners and bounded away polytopes, which in my opinion form central technical elements of the work, are not even defined in the main body. My other criticisms are rather minor and can be found in the comments section.
# Suggestions for Improvement
In my view, it is important that the authors include more details about their technical contributions in the main paper. Given that the conference organisers relaxed the 4 page restriction on submissions I would suggest extending the main body of the paper to 6 pages to accommodate this.
In particular, I would provide the following soft-guidance to the authors:
- Move most of Appendix B to the main body. In my view, this where most of the “meat” of the paper is.
I would keep Definition B.3 in the appendices, as I feel it is safe to assume the reader is familiar with this result (and if they aren’t then they can look it up in the Appendix). Similarly, I would keep Lemma B.9 in the appendix as it is an intermediate result.
- Move some of the content in Appendix C to the main body.
(1) The content regarding computation of the Caratheodory decomposition can be left in the appendix, as this is mostly based on the existing work of Pangeas (2023). Though the fact that you can efficiently find this decomposition should be mentioned in the main body.
(2) The content regarding construction of spanners for DAGs should be included in the main body. (3) Algorithm 3 can probably be left in the appendices since it is very similar to Algorithm 2.
I would probably keep the proof of Theorem C.9 in the appendix just to cut down the length of the main body.
(4) Keep the remaining appendices in the appendix to stop the main body from being too long.

Of course, the guidance above is just my recommendation. The authors know their technical contributions far better than me and are better placed to decide which technical elements are necessary/interesting to include in the main body.

# Other Minor Comments
- I would change each of the “No”s in Table 1 to “unknown” unless there is an explicit counterexample showing that these bandit algorithms don’t converge to Nash equilibria.
- “table 1” should be capitalised on line 106/107.
- In the abstract you may want to briefly mention that your algorithm uses Barycentric spanners. When I originally read the abstract it was unclear to me why you were mentioning Barycentric spanners. This was only clarified properly when I read the appendices.
- Caratheodory is misspelled on line 668-670.
- There are Latex formatting issues with Lemma C.2 as well as on line 712-713.
- I do did not read through Appendices A, D, E, F, G, H, I in great detail. However, the methodology of the authors seems sound from a technical standpoint.